# Adaptive Stochastic Variance Reduction for Non-convex Finite-Sum Minimization

**Ali Kavis**[*]
LIONS, EPFL
ali.kavis@epfl.ch

**Stratis Skoulakis**[*]
LIONS, EPFL
efstratios.skoulakis@epfl.ch

**Kimon Antonakopoulos**
LIONS, EPFL
kimon.antonakopoulos@epfl.ch

**Leello Tadesse Dadi**
LIONS, EPFL
leello.dadi@epfl.ch

**Volkan Cevher**
LIONS, EPFL
volkan.cevher@epfl.ch

## Abstract

We propose an adaptive variance-reduction method, called ADASPIDER, for minimization of $L$-smooth, non-convex functions with a finite-sum structure. In essence, ADASPIDER combines an ADAGRAD-inspired [16, 40], but a fairly distinct, adaptive step-size schedule with the recursive *stochastic path integrated estimator* proposed in Fang et al. [19]. To our knowledge, ADASPIDER is the first parameter-free non-convex variance-reduction method in the sense that it does not require the knowledge of problem-dependent parameters, such as smoothness constant $L$, target accuracy $\epsilon$ or any bound on gradient norms. In doing so, we are able to compute an $\epsilon$-stationary point with $\tilde{O}\left(n + \sqrt{n}/\epsilon^2\right)$ oracle-calls, which matches the respective lower bound up to logarithmic factors.

## 1   Introduction

This paper studies smooth, non-convex minimization problems with the following finite-sum structure:

$$\min_{x\in\mathbb{R}^d} f(x) := \frac{1}{n}\sum_{i=1}^n f_i(x), \tag{Prob}$$

where each component function $f_i : \mathbb{R}^d \to \mathbb{R}$ is $L$-smooth and is possibly non-convex, and we further assume $f$ is also non-convex. We seek to find an $\epsilon$-approximate first-order stationary point $\hat{x}$ of $f$, such that $\|\nabla f(\hat{x})\| \leq \epsilon$, where $\epsilon > 0$ is the accuracy of the desired solution.

This structure captures many interesting learning problems from empirical risk minimization to training of neural networks. *First-order methods* have been the standard choice for solving (Prob), due to their efficiency and favorable practical behavior. In that regard, while gradient descent (GD) requires $O(n/\epsilon^2)$ gradient computations, stochastic gradient descent (SGD) requires $O(1/\epsilon^4)$ overall gradient computations. In many interesting machine learning applications $n$ tends to be large, e.g., training a neural network for image classification with very big image datasets [14], hence SGD typically leads to better practical performance.

---

[*]Joint first authors

36th Conference on Neural Information Processing Systems (NeurIPS 2022).

To leverage the best of both regimes, GD and SGD, the so-called variance reduction (VR) framework combines the *faster convergence rate* of GD with the *low per-iteration complexity* of SGD. Originally proposed for solving strongly-convex problems [24, 13, 42], variance reduction frameworks essentially generate low-variance gradient estimates by maintaining a balance between periodic full gradient computations and stochastic (mini-batch) gradients. VR methods and their theoretical behavior for *convex problems* have been well-studied under various problem setups and assumptions, including $\mu$-strongly convex functions with $O(n + (L/\mu)\log(1/\epsilon))$ complexity [24, 42, 13]; $\mu$-strongly convex functions with accelerated $O(n + \sqrt{L/\mu}\log(1/\epsilon))$ complexity [2, 29, 47] and smooth, convex functions with $\tilde{O}(n+1/\epsilon)$ complexity [58, 47, 15]. For non-convex minimization, earlier attempts extended the existing VR frameworks, achieving the first rates of order $O(n+n^{2/3}/\epsilon^2)$ with sub-optimal dependence on $n$ [45, 55, 1, 35]. The most recent non-convex VR methods [18, 51, 36, 44, 37] close this gap and achieve the optimal gradient oracle complexity of $O(n + \sqrt{n}/\epsilon^2)$ [18].

**Adaptivity and First-order Optimization**

The selection of the step-size is of great importance in both the theoretical and practical performance of first-order methods, including the aforementioned VR methods. In the case of $L$-smooth minimization, first-order methods need the knowledge of $L$ so as to adequately select their step-size [41], otherwise the method is not guaranteed to be convergent and might even diverge [15, 38]. To elucidate, classical analysis relies on the (expected) descent property and guarantees that the algorithm monotonically makes progress every iteration. To enforce this property everywhere on the optimization landscape, one needs to pick the step-size as $\gamma_t \leq O(1/L)$, which restricts the step length of the algorithm with respect to the worst-case constant $L$. On the other hand, estimating the smoothness constant for an objective of interest, such as neural networks, is a very hard task [21]. At the same time, using crude bounds on the smoothness constant leads to very small step-sizes and consequently to poorer convergence. In practice the step-size is tuned through an empirical search over a range of hand-picked values that adds a considerable computational overhead and burden. In order to alleviate the burden of tuning process, we need step-sizes that adjust in accordance with the optimization path.

A popular line of research studies first-order methods that *adaptively* select their step-size by taking advantage of the previously produced point. In many settings of interest, these *adaptive methods* are able to guarantee optimal convergence rates without requiring the knowledge of the smoothness constant $L$ while they often admit superior empirical performance due to their ability to decrease the step-size according to the local geometry of the objective function. Inspired by AdaGrad introduced in the concurrent seminal works of [16] [39], a recent line of works [31, 26, 25, 17] propose adaptive gradient methods that given access to *noiseless gradient-estimates* achieve accelerated rates in the case of $L$-smooth convex minimization without requiring the knowledge of smoothness constant $L$. Similarly, Ene et al. [17] and Antonakopoulos et al. [5] propose adaptive methods with optimal convergence rates for monotone variational inequalities while Antonakopoulos et al. [6] provide adaptive methods for monotone variational inequalities assuming access to *relative noise gradient-estimates*. Hsieh et al. [23] and Vu et al. [50] study the convergence properties of adaptive first-order methods for routing and generic games.

***Adaptive non-convex methods for General Noise*** Related to our work is a recent line of papers studying adaptive first-order methods under the *general noise model*. In this setting, a method is assumed to access unbiased stochastic estimate of the gradient with bounded variance. This is a more general setting than finite-sum optimization that comes with worse lower bounds, i.e. $\Omega(1/\epsilon^4)$[2] gradient-estimates are needed so as to compute an $\epsilon$-stationary point. A recent line of works study adaptive first-order methods that are able to achieve near-optimal oracle-complexity while being oblivious to the smoothness constant $L$ and the variance of the estimator [52, 20, 34, 27]. For example Ward et al. [52] established that the adaptive method called AdaGrad-Norm is able to achieve $\tilde{O}(1/\epsilon^4)$ gradient-complexity in the general noise model. In their recent work, Faw et al. [20] significantly extended the results of Ward et al. [52] by showing that AdaGrad-Norm achieves the same rates even in case the gradient admits unbounded norm (a restrictive assumption in [52]) while their result persists even if the variance increases with the gradient norm. In the slightly more restrictive setting at which the objective function $f(x) := \mathbb{E}_{\xi \sim \mathcal{D}} g(x, \xi)$ where each estimate $g(x, \xi)$ is $L$-smooth with respect to $x$ for all $\xi$, Levy et al. [32] proposed an adaptive method called STORM++ that achieves $O(1/\epsilon^3)$ gradient-complexity and that removes the requirement of the knowledge on

---

[2]We remark that in the case of finite-sum minimization there exist variance reduction methods with $O(n + \sqrt{n}/\epsilon^2)$ gradient complexity [19, 51].

Table 1: In the following table we present the gradient computation complexity of the existing non-adaptive and adaptive variance reduction methods for both convex and non-convex finite-sum minimization. Since for there are multiple non-adaptive VR methods, we present the earliest-proposed method matching up to logarithmic factors the respective lower bounds.

| $f(x)$ | Non-Adaptive VR | Adaptive VR | Complexity Lower Bound |
|---|---|---|---|
| convex ($\epsilon$-opt. solution) | $\tilde{O}\left(n + \sqrt{\frac{n}{\epsilon}}\right)$ [28] | $\tilde{O}\left(n + \sqrt{\frac{n}{\epsilon}}\right)$ [38] | $\Omega(n + \sqrt{\frac{n}{\epsilon}})$ [53] |
| convex ($\epsilon$-opt. solution) | $\tilde{O}\left(n + \frac{1}{\epsilon}\right)$ [57] | $\tilde{O}\left(n + \frac{1}{\epsilon}\right)$ [15] | $\Omega(n + \sqrt{\frac{n}{\epsilon}})$ [53] |
| non-convex ($\epsilon$-stat. point) | $O\left(n + \frac{\sqrt{n}}{\epsilon^2}\right)$ [19] | $\tilde{O}\left(n + \frac{\sqrt{n}}{\epsilon^2}\right)$ This work | $\Omega\left(n + \frac{\sqrt{n}}{\epsilon^2}\right)$ [19] |

problem parameters (e.g., smoothness constant, absolute bounds on gradient norms) that the original STORM method proposed by Cutkosky et al. [12] requires. The latter gradient-complexity matches the $\Omega(1/\epsilon^3)$ lower bound of Arjevani et al. [7].

**Adaptivity and Finite-Sum minimization**

In parallel with what we discussed earlier, existing variance-reduction methods (VR) crucially need to know the smoothness constant $L$ to select their step-size appropriately to guarantee their convergence. To this end, the following natural question arises

*Can we design **adaptive** VR methods that achieve the optimal gradient computation complexity?*

Li et al. [33] and Tan et al. [49] were the first to propose adaptive variance-reduction methods by using the Barzilai-Borwein step-size [8]. Despite their promising empirical performance, these methods do not admit formal convergence guarantees. When the objective function $f$ in (Prob) is convex, [15] recently proposed an adaptive VR method requiring $O(n + 1/\epsilon)$ gradient computation while, shortly after, [38] proposed an *accelerated* adaptive VR method requiring $O(n + \sqrt{n}/\sqrt{\epsilon})$ gradient computations.

To the best of our knowledge, there is no adaptive VR method in the case where $f$ is *non-convex*. We remark that $f$ being non-convex captures the most interesting settings such as minimizing the empirical loss of deep neural network where each $f_i$ stands for the loss with respect to $i$-th data point and thus is a non-convex function in the parameters of the neural architecture. Through this particular example, we could motivate adaptive VR methods in two fronts: first, even estimating the smoothness constant $L$ of a deep neural network is prohibitive [21], and at the same time, the parameter $n$ in (Prob) equals the number of data samples, which can be very large in practice and is prohibitive for the use of deterministic methods.

**Contribution and Techniques** In this work we present an adaptive VR method, called ADASPIDER, that converges to an $\epsilon$-*stationary point* for (Prob) by using $\tilde{O}(n + \sqrt{n}L^2/\epsilon^2)$ gradient computations. Our gradient complexity bound matches the existing lower bounds up to logarithmic factors [19].

ADASPIDER combines an adaptive step-size schedule in the lines proposed by ADAGRAD [16] with the variance-reduction mechanism based on the *stochastic path integrated differential estimator* of the SPIDER algorithm [19]. More precisely, ADASPIDER selects the step-size by aggregating the norm of its recursive estimator, while following a single-loop structure as in Fang et al. [19].

Our contributions and techniques can be summarized as follows:

- To our knowledge, ADASPIDER is *the first* parameter-free method in the sense that it is both *accuracy-independent* and is oblivious to the knowledge of any problem parameters including $L$. Moreover, $\epsilon$-independence enables us to provide *any-iterate* guarantees. While SPIDER needs both $\epsilon$ and $L$ to set its step-size as $\min\left(\frac{\epsilon}{L\sqrt{n}\|\nabla_t\|}, \frac{1}{2\sqrt{n}L}\right)$ to achieve optimal gradient complexity [19], all other existing non-convex methods must know at least the value of $L$ in order to guarantee convergence [4, 51].

- We introduce a novel step-size schedule $\gamma_t := n^{-1/4} \left( \sqrt{n} + \sum_{s=0}^{t} \|\nabla_s\|^2 \right)^{-1/2}$ where $\nabla_s$ is the recursive variance-reduced estimator at round $s$. By identifying a unique additive/multiplicative form for integrating $n$, we manage to achieve optimal dependence on the number of components. We note that Adaspider can be viewed as SPIDER with the step-size of AdaGrad-Norm [20, 52, 48, 43] were the parameters are respectively selected as $\eta := n^{1/4}$ and $b_0^2 := \sqrt{n}$ [20].

- We show how to combine the above adaptive step-size schedule with the recursive SPIDER estimator in order to ensure that the *average variance* $\frac{1}{T} \sum_{t=0}^{T-1} \mathbb{E}\left[\|\nabla_t - \nabla f(x_t)\|\right]$ decays at a rate $\tilde{O}\left(n^{1/4}/\sqrt{T}\right)$. This might be of independent interest for other variance reduction techniques.

We follow a novel technical path that uses the adaptivity of the step-size to bound the overall variance of the process. This fact differentiates our approach from the previous adaptive and non adaptive VR approaches (see Section 4 for further details) and provides us a surprisingly concise analysis.

*Remark* 1. Our convergence results do not require bounded gradients that is typically a restrictive assumption that the analysis of the adaptive methods for stochastic optimization require. We overcome this obstacle by using the fact $\|x_t - x_{t-1}\| \leq 1$ (due to the step-size selection) and thus $\|\nabla f(x_t) - \nabla f(x_{t-1})\| \leq L\|x_t - x_{t-1}\| \leq L$. The latter leads to the following upper bound on the gradient norm, $\|\nabla f(x_t)\| \leq LT + \|\nabla f(x_0)\|$ that however leads to only a logarithmic overhead in the final bound (see Lemma 2). A similar idea is used by Faw et al. [20] (Lemma 2) in order to remove the bounded gradient assumption on the convergence rates of AdaGrad-Norm under general noise.

## 2 Setup and Preliminaries

During the whole of this manuscript, we consider that the non-convex objective function $f : \mathbb{R}^d \mapsto \mathbb{R}$ possesses a finite-sum structure

$$f(x) = \frac{1}{n} \sum_{i=1}^{n} f_i(x)$$

where each component function $f_i$ is *L-smooth* (or alternatively has *L-Lipschitz gradient*) and (possibly) non-convex. To quantify the performance of our algorithm within the context of non-convex minimization, we want to find an $\epsilon$-*first order stationary point* $\hat{x} \in \mathbb{R}^d$ such that

$$\|\nabla f(\hat{x})\| \leq \epsilon.$$

For notational simplicity we define $\|\cdot\|$ as the Euclidean norm. Then, we say that a continuously differentiable function $f$ is *L-smooth* if

$$\|\nabla f(x) - \nabla f(y)\| \leq L\|x - y\|, \tag{2.1}$$

which admits the following equivalent form,

$$f(x) \leq f(y) + \nabla f(y)^\top (x - y) + \frac{L}{2}\|x - y\|^2 \quad \text{for all } x, y \in \mathbb{R}^d. \tag{2.2}$$

Observe that smoothness of each component immediately suggests that objective $f$ is $L$-smooth itself. Since we are studying randomized algorithms for finite-sum minimization problems, we do not consider any variance bounds on the gradients of components. We only assume that we have access to an oracle which returns the gradient of individual components when queried.

## 3 Adaptive SPIDER algorithm and convergence results

In this section, we present our adaptive variance reduction method, called ADASPIDER (Algorithm 1) which exploits the variance reduction properties of the *stochastic path integrated differential estimator* proposed in [19] while combining it with an AdaGrad-type step-size construction [16]. Unlike the original SPIDER method [19], our algorithm admits anytime guarantees, i.e., we don't need to specify the accuracy $\epsilon$ a priori. Additionally, our algorithm does not need to know the smoothness parameter $L$ and guarantees convergence without any tuning procedure.

---

**Algorithm 1** Adaptive SPIDER (ADASPIDER)

---

**Input:** $x_0 \in \mathbb{R}^d, \beta_0 > 0, G_0 > 0$

1: $G \leftarrow 0$
2: **for** $t = 0, ..., T - 1$ **do**
3:     **if** $t \bmod n = 0$ **then**
4:         $\nabla_t \leftarrow \nabla f(x_t)$
5:     **else**
6:         pick $i_t \in \{1, \ldots, n\}$ uniformly at random
7:         $\nabla_t \leftarrow \nabla f_{i_t}(x_t) - \nabla f_{i_t}(x_{t-1}) + \nabla_{t-1}$
8:     **end if**
9:     $\gamma_t \leftarrow 1 / \left( n^{1/4} \beta_0 \sqrt{n^{1/2} G_0^2 + \sum_{s=0}^t \|\nabla_s\|^2} \right)$
10:    $x_{t+1} \leftarrow x_t - \gamma_t \cdot \nabla_t$
11: **end for**
12: **return** uniformly at random $\{x_0, \ldots, x_{T-1}\}$.

---

*Remark* 2. In Algorithm 1 the units of $G_0$ are the same with the units of $\nabla f(x_t)$ i.e. $f/x$ while the units of $\beta_0$ are $x^{-1}$. The latter is important so that the step-size $\gamma_t$ takes the right units i.e. $x^2/f$.

As Algorithm 1 indicates, ADASPIDER performs a full-gradient computation every $n$ iterations while in the rest iterations updates the variance-reduced gradient estimator in a recursive manner, $\nabla_t \leftarrow \nabla f_{i_t}(x_t) - \nabla f_{i_t}(x_{t-1}) + \nabla_{t-1}$. The adaptive nature of ADASPIDER comes from the selection of the step-size at Step 9 that only depends on the norms of estimates produced by the algorithm in the previous steps.

Before presenting the formal convergence guarantees of ADASPIDER (stated in Theorem 1), we present the cornerstone idea behind its design and motivate the analysis for controlling the overall variance of the process through the *adaptivity of the step-size*. This conceptual novelty differentiates our work form the previous adaptive VR methods [15, 38] at which the adaptivity of the step-size is only used for adapting to the smoothness constant $L$, and their constructions come with additional challenges in bounding the variance. As a result, the following challenge is the first to be tackled by the design of a VR method.

**Challenge 1.** *Does the average variance of the estimator,* $\frac{1}{T} \sum_{t=0}^{T-1} \mathbb{E} [\|\nabla_t - \nabla f(x_t)\|]$, *diminishes at a sufficiently fast rate?*

Up next we explain why combining the variance-reduction estimator of Step 7 with the adaptive step-size of Step 9 provides a surprisingly concise answer to Challenge 1. We remark that SPIDER is able to control the variance at any iterations by choosing $\gamma_t := \min(\frac{\epsilon}{L\sqrt{n}\|\nabla_t\|}, \frac{1}{2\sqrt{n}L})$ as step-size. The latter enforces the method to make tiny steps, $\|x_t - x_{t-1}\| \leq \epsilon/L\sqrt{n}$ which results in $\epsilon$-bounded variance at any iteration. The latter proposed SPIDERBOOST [51] provides the same gradient-complexity bounds with SPIDER but through the *accuracy-independent* step-size $\gamma = 1/L$. SPIDERBOOST handles Challenge 1 by using a dense gradient-computations schedule[3] combined with amortization arguments based on the descent inequality (this is why the knowledge of $L$ is necessary in its analysis). We remark that ADASPIDER, despite being oblivious to $L$ and accuracy $\epsilon$, admits a significantly simpler analysis by exploiting the adaptability of its step-size.

In the rest of the section we present our approach to Challenge 1 and we conclude the section with Theorem 1 stating the formal convergence guarantees of ADASPIDER.

**Handling the variance with adaptive step-size** We start with the following variance aggregation lemma that is folklore in (VR) literature (e.g. [56]).

**Lemma 1.** *Define the gradient estimator at point $x$ as $\nabla_x := \nabla f_i(x) - \nabla f_i(y) + \nabla_y$ where $i$ is sampled uniformly at random from $\{1, \ldots, n\}$. Then,*

$$\mathbb{E} \left[ \|\nabla_x - \nabla f(x)\|^2 \right] \leq L^2 \|x - y\|^2 + \mathbb{E} \left[ \|\nabla_y - \nabla f(y)\|^2 \right]$$

---

[3]ADASPIDER computes a full-gradient every $\sqrt{n}$ steps and at the intermediate steps uses batches of size $\sqrt{n}$.

Now, let us apply Lemma 1 on SPIDER estimator, $\nabla_t := \nabla f_{i_t}(x_t) - \nabla f_{i_{t-1}}(x_t) + \nabla_{t-1}$ to measure its variance at step $x_t$.

$$
\begin{aligned}
\mathbb{E}\left[\|\nabla_t - \nabla f(x_t)\|^2\right] &\leq L^2 \mathbb{E}\left[\|x_t - x_{t-1}\|^2\right] + \mathbb{E}\left[\|\nabla_{t-1} - \nabla f(x_{t-1})\|^2\right] \\
&\leq L^2 \mathbb{E}\left[\gamma_{t-1}^2\|\nabla_{t-1}\|^2\right] + \mathbb{E}\left[\|\nabla_{t-1} - \nabla f(x_{t-1})\|^2\right] \\
&\leq L^2 \mathbb{E}\left[\gamma_{t-1}^2\|\nabla_{t-1}\|^2\right] + \ldots + \mathbb{E}\left[\|\nabla_{t-(t \bmod n)} - \nabla f(x_{t-(t \bmod n)})\|^2\right] \\
&= \sum_{\tau=t-(t \bmod n)+1}^{t-1} L^2 \mathbb{E}\left[\gamma_\tau^2 \cdot \|\nabla_\tau\|^2\right]
\end{aligned}
$$

where the last equality follows by the fact $\mathbb{E}\left[\|\nabla_{t-(t \bmod n)} - \nabla f(x_{t-(t \bmod n)})\|^2\right] = 0$ since Algorithm 1 performs a full-gradient computations for every $t$ with $t \bmod n = 0$ (Step 3 of Algorithm 1). By telescoping the summation we get,

$$
\sum_{t=0}^{T-1} \mathbb{E}\left[\|\nabla_t - \nabla f(x_t)\|^2\right] \leq \sum_{t=0}^{T-1} \sum_{\tau=t-(t \bmod n)+1}^{t-1} L^2 \mathbb{E}\left[\gamma_\tau^2 \cdot \|\nabla_\tau\|^2\right] \leq L^2 n \cdot \sum_{t=0}^{T-1} \mathbb{E}\left[\gamma_t^2 \cdot \|\nabla_t\|^2\right]
$$

where the $n$ factor on the right-hand side is due to the fact that each term $\mathbb{E}\left[\gamma_t^2\|\nabla_t\|^2\right]$ appears at most $n$ times in the total summation. To this end, using the structure of the *stochastic path integrated differential estimator* we have been able to bound the overall variance of the process as follows,

$$
\sum_{t=0}^{T-1} \mathbb{E}\left[\|\nabla_t - \nabla f(x_t)\|^2\right] \leq L^2 n \cdot \sum_{t=0}^{T-1} \mathbb{E}\left[\gamma_t^2 \cdot \|\nabla_t\|^2\right]
$$

However it is not clear at all why the above bound is helpful. At this point the adaptive selection of the step-size (Step 9 in Algorithm 1) comes into play by providing the following surprisingly simple answer,

$$
\begin{aligned}
\sum_{t=0}^{T-1} \mathbb{E}\left[\|\nabla_t - \nabla f(x_t)\|^2\right] &\leq L^2 n \, \mathbb{E}\left[\sum_{t=0}^{T-1} \gamma_t^2 \cdot \|\nabla_t\|^2\right] \\
= \frac{L^2 \sqrt{n}}{\beta_0^2} \, \mathbb{E}\left[\sum_{t=0}^{T-1} \frac{\|\nabla_t\|^2/G_0^2}{\sqrt{n} + \sum_{s=0}^{t}\|\nabla_s\|^2/G_0^2}\right] &\leq \frac{L^2 \sqrt{n}}{\beta_0^2} \log\left(1 + \mathbb{E}\left[\sum_{t=0}^{T-1}\|\nabla_t\|^2/G_0^2\right]\right)
\end{aligned}
$$

where the last inequality comes from Lemma 7. To finalize the bound, we require the following expression that follows by the fact that $\gamma_t \leq 1/\|\nabla_t\|$ and thus $\|x_t - x_{t-1}\| \leq 1$.

**Lemma 2.** *Let $x_0, x_1, \ldots, x_T$ the points produced by Algorithm 1. Then,*

$$
\sum_{t=0}^{T-1}\|\nabla_t\|^2 \leq \mathcal{O}\left(n^2 T^3 \cdot \left(\frac{L^2}{\beta_0^2} + \|\nabla f(x_0)\|^2\right)\right)
$$

In simple terms, Lemma 2 helps us avoid the *bounded gradient norm* assumption that is common among adaptive non-convex methods. As a result, ADASPIDER admits the following cumulative variance bound,

$$
\sum_{t=0}^{T-1} \mathbb{E}\left[\|\nabla_t - \nabla f(x_t)\|^2\right] \leq O\left(\frac{L^2\sqrt{n}}{\beta_0^2}\log\left(1 + nT \cdot \left(\frac{L}{\beta_0 G_0} + \frac{\|\nabla f(x_0)\|}{G_0}\right)\right)\right). \tag{3.1}
$$

*Remark* 3. To this end one might notice that using a more aggressive $n$ dependence on $\gamma_t$ leads to smaller variance of the estimator which is obviously favorable. However more aggressive dependence on $n$ leads to smaller step-sizes and thus to sub-optimal overall gradient-computation complexity. In Section 4, we explain why the optimal way to inject the $n$ dependence into the step-size is through the simultaneous multiplicative/additive way described in Step 9 of ADASPIDER that may seem unintuitive on the first sight.

We will conclude this discussion with a complementary remark on the interplay between our adaptive step-size and the convergence rate. As we demonstrated in Eq. (3.1), using a data-adaptive step-size leads to a decreasing variance bound *in an amortized sense* as opposed to *any iterate* variance bound of SPIDER. The trade-off in our favor is the parameter-free step-size that is independent of $\epsilon$ and $L$. For a fair exposition of our results, notice that the aforementioned advantages of an adaptive step-size comes at an additional $\log(T)$ term in our final bound due to Eq. (3.1). This has a negligible effect on the convergence as even in the large iteration regime when $T$ is in the order billions, it amounts to a small constant factor.

We conclude the section with Theorem 1 that formally establishes the convergence rate of ADASPI-DER. The proof of Theorem 1 is deferred for the next section.

**Theorem 1.** *Let $x_0, x_1, \ldots, x_{T-1}$ be the sequence of points produced by Algorithm 1 in case $f(\cdot)$ is $L$-smooth. Let us also define $\Delta_0 := f(x_0) - f^*$. Then,*

$$\frac{1}{T} \sum_{t=0}^{T-1} \mathbb{E}\left[\|\nabla f(x_t)\|\right] \leq O\left(n^{1/4} \cdot \frac{\Theta}{\sqrt{T}} \cdot \log\left(1 + nT \cdot \left(\frac{L}{\beta_0 G_0} + \frac{\|\nabla f(x_0)\|}{G_0}\right)\right)\right)$$

*where $\Theta = \Delta_0 \cdot \beta_0 + G_0 + L/\beta_0 + L^2/(\beta_0^2 G_0)$. Overall, Algorithm 1 with $\beta_0 := 1$ and $G_0 := 1$ needs at most $\tilde{O}\left(n + \sqrt{n} \cdot \frac{\Delta_0^2 + L^4}{\epsilon^2}\right)$ oracle calls to reach an $\epsilon$-stationary point.*

## 4   Sketch of Proof of Theorem 1

In this section we present the key steps for proving Theorem 1. We first use the triangle inequality to derive,

$$\sum_{t=0}^{T-1} \mathbb{E}\left[\|\nabla f(x_t)\|\right] \leq \sum_{t=0}^{T-1} \mathbb{E}\left[\|\nabla_t - \nabla f(x_t)\|\right] + \sum_{t=0}^{T-1} \mathbb{E}\left[\|\nabla_t\|\right]$$

We have previously discussed how to bound the first term in Section 3. More precisely, by the Jensen's inequality and the arguments presented in Section 3, we obtain the following variance bound.

**Lemma 3.** *Let $x_0, x_1 \ldots, x_T$ the sequence of points produced by Algorithm 1. Then,*

$$\sum_{t=0}^{T-1} \mathbb{E}\left[\|\nabla_t - \nabla f(x_t)\|^2\right] \leq O\left(\frac{Ln^{1/4}}{\beta_0}\sqrt{\log\left(1 + nT \cdot \left(\frac{L}{\beta_0 G_0} + \frac{\|\nabla f(x_0)\|}{G_0}\right)\right)}\right).$$

We continue with presenting how to treat the term $\sum_{t=0}^{T-1} \mathbb{E}\left[\|\nabla_t\|\right]$. By the smoothness of the function and through a telescopic summation one can easily establish the following bound,

$$\mathbb{E}\left[\sum_{t=0}^{T-1} \gamma_t \cdot \|\nabla_t\|^2\right] \leq 2(f(x_0) - f^*) + L \cdot \mathbb{E}\left[\sum_{t=0}^{T-1} \gamma_t^2 \cdot \|\nabla_t\|^2\right] + \mathbb{E}\left[\sum_{t=0}^{T-1} \gamma_t \cdot \|\nabla f(x_t) - \nabla_t\|^2\right]$$

As we explained in Section 3, the term $\mathbb{E}\left[\sum_{t=0}^{T-1} \gamma_t^2 \cdot \|\nabla_t\|^2\right]$ can be upper bounded by the adapt-ability of the step-size $\gamma_t$. At the same time, using the adaptability of $\gamma_t$ we are able to establish that $\sum_{t=0}^{T-1} \mathbb{E}\left[\|\nabla_t\|\right]$ is at most $n^{1/4}\sqrt{T} \cdot \mathbb{E}\left[\sum_{t=0}^{T-1} \gamma_t \cdot \|\nabla_t\|^2\right]$. All the above are formally stated and established in Lemma 4 the proof of which is deferred to the appendix.

**Lemma 4.** *Let $x_0, x_1, \ldots, x_{T-1}$ the sequence of points produced by Algorithm 1 and $\Delta_0 := f(x_0) - f^*$. Then,*

$$\mathbb{E}\left[\sum_{t=0}^{T-1} \|\nabla_t\|\right] \leq \tilde{\mathcal{O}}\left(\Delta_0 \cdot \beta_0 + G_0 + \frac{L}{\beta_0} + \mathbb{E}\left[\sum_{t=0}^{T-1} \gamma_t \|\nabla f(x_t) - \nabla_t\|^2\right]\right) n^{1/4}\sqrt{T}.$$

Due to the use of the adaptive step-size $\gamma_t$, the *estimator's error* and the step-size itself are dependent random variables, meaning that the weighted variance term $\mathbb{E}\left[\sum_{t=0}^{T-1} \gamma_t \|\nabla f(x_t) - \nabla_t\|^2\right]$ cannot be handled by Lemma 1. To overcome the latter, we use the monotonic behavior of the step-size $\gamma_t$ to establish the following refinement of Lemma 1.

**Lemma 5.** *Let $x_0, x_1, \ldots$ the sequence of points produced by Algorithm 1. Then,*

$$\mathbb{E}\left[\sum_{t=0}^{T-1} \gamma_t \cdot \|\nabla_t - \nabla f(x_t)\|^2\right] \leq L^2 n \cdot \mathbb{E}\left[\sum_{t=0}^{T-1} \gamma_t^3 \cdot \|\nabla_t\|^2\right]$$

To this end we are ready to summarize the importance of simultaneous additive/multiplicative dependence of $\gamma_t$ in Step 9 of Algorithm 1. This selection permits us to do achieve two *orthogonal goals* at the same time,

- Bounding the variance of the process, $\mathbb{E}\left[\sum_{t=0}^{T-1} \|\nabla_t - \nabla f(x_t)\|\right] \leq \tilde{O}\left(n^{1/4}\sqrt{T}\right)$ (see Section 3 and Lemma 3).

- Bounding the sum, $\mathbb{E}\left[\sum_{t=0}^{T-1} \|\nabla_t\|\right] \leq \tilde{O}\left(n^{5/4}\sqrt{T} \cdot \mathbb{E}\left[\sum_{t=0}^{T-1} \gamma_t^3 \|\nabla_t\|^2\right]\right)$ (see Lemma 4 and Lemma 5).

The third important thing that the selection of $\gamma_t$ does is that it permits us to upper bound the term $\tilde{O}\left(n^{5/4} \mathbb{E}\left[\sum_{t=0}^{T-1} \gamma_t^3 \|\nabla_t\|^2\right]\right)$ by $\tilde{O}\left(n^{1/4}\right)$. The proof of the latter upper bound can be found in the proof of Theorem 1 that due to lack of space is deferred to appendix. It is interesting that all the above three different purposes can be handled by selecting $\gamma_t = n^{-1/4}\beta_0(\sqrt{n} \cdot G_0^2 + \sum_{s=0}^{t} \|\nabla_s\|^2)^{-1/2}$.

## 5 Experiments

We complement our theoretical findings with an evaluation of the numerical performance of the algorithm under different experimental setups. We aim to highlight the sample complexity improvements over simple stochastic methods, while displaying the advantages of adaptive step-size strategies. For that purpose we design two setups; first, we consider the minimization of a convex loss with a non-convex regularizer in the sense of Wang et al. [51] and in a second part we consider an image classification task with neural networks.

### 5.1 Convex loss with a non-convex regularizer

We consider the following problem: $\min_{x \in \mathbb{R}^d} \frac{1}{n} \sum_{i=1}^n \ell(x, (a_i, b_i)) + \lambda g(x)$ where $\ell(x, (a_i, b_i))$ is the loss with respect to the decision variable/weights $x$ with $(a_i, b_i)$ denoting the (feature vector, label) pair. We select $g = \sum_{i=1}^d \frac{x_i^2}{1+x_i^2}$, similar to Wang et al. [51], where the subscript denotes the corresponding dimension of $x$. We compare ADASPIDER against the original SPIDER, SPIDERBOOST, SVRG, ADASVRG and two non-VR methods, SGD and ADAGRAD. We picked two datasets from LibSVM, namely a1a, mushrooms. We initialize each algorithm from the same point and repeat the experiments 5 times, then report the mean convergence with standard deviation as the shaded region around the mean curves. We tune the algorithms by executing a parameter sweep for their initial step-size over an interval of values which are exponentially scaled as $\left\{10^{-3}, 10^{-2}, ..., 10^2, 10^3\right\}$. After tuning the algorithms on one dataset, we run them with the same parameters for the others.

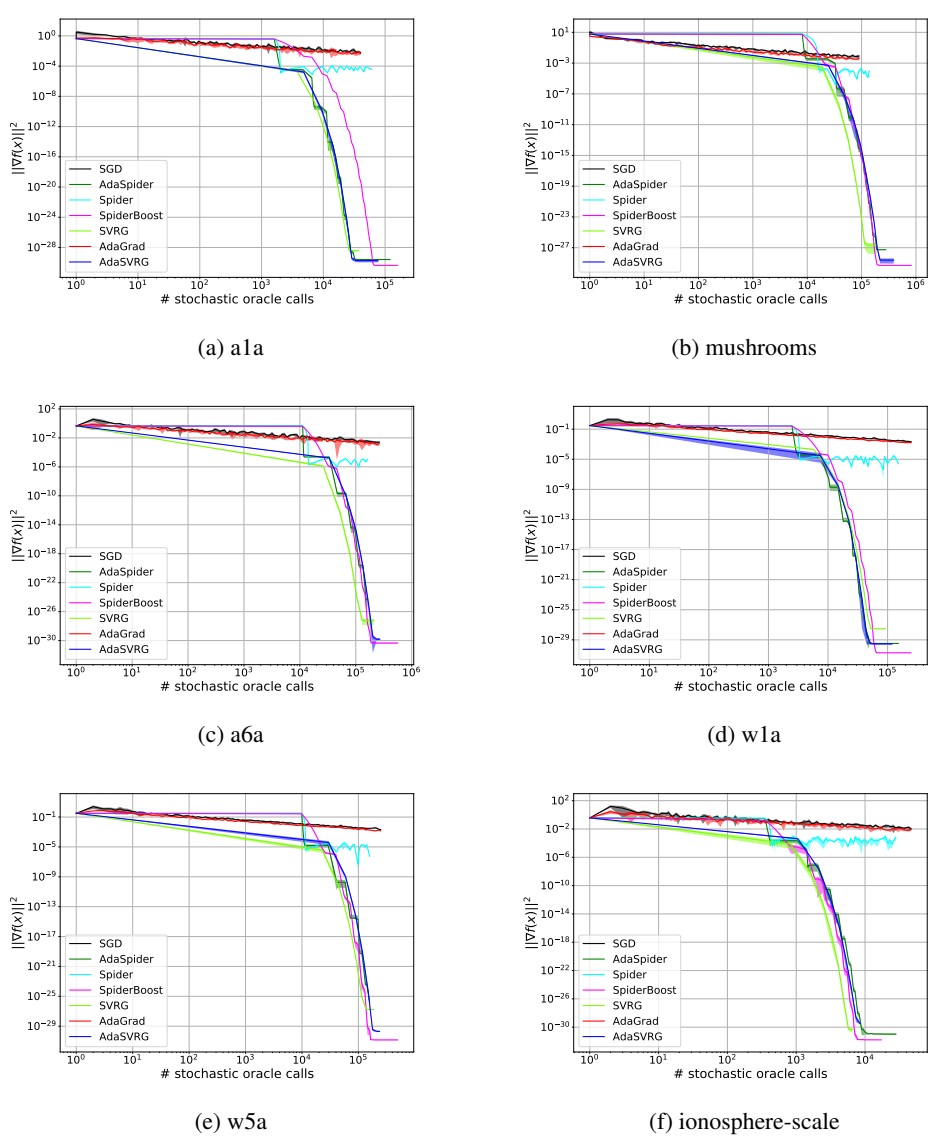

Figure 1: Logistic regression with non-convex regularizer on LibSVM datasets

First, we clearly observe the difference between SGD & ADAGRAD, and the rest of the pack, which demonstrates the superior sample complexity of VR methods in general. Among VR algorithms, there does not seem to be any concrete differences with similar convergence, except for SPIDER. The performance of ADASPIDER is on par with other VR methods, and superior to SPIDER. The unexpected behavior of SPIDER algorithm has previously been documented in Wang et al. [51]. From a technical point of view, this behavior is predominantly due to the *accuracy dependence* in the step-size, making the step-size unusually small. We had to run SPIDER beyond its prescribed setting and tune the step-size with a large initial value to make sure the algorithm makes observable progress.

## 5.2 Experiments with neural networks

In our second setup, we train neural networks with our variance reduction scheme. Our focus is on standard image classification tasks trained with the cross entropy loss [9, 10]. Denoting by $C$ the number of classes, the considered datasets in this section consist of $n$ pairs $(a_i, b_i)$ where $a_i$ is a vectorized image and $b_i \in \mathbb{R}^C$ is a one-hot encoded class label. A neural network is parameterized with weights $x \in \mathbb{R}^d$ and its output on is denoted net(x, a) $\in \mathbb{R}^C$,

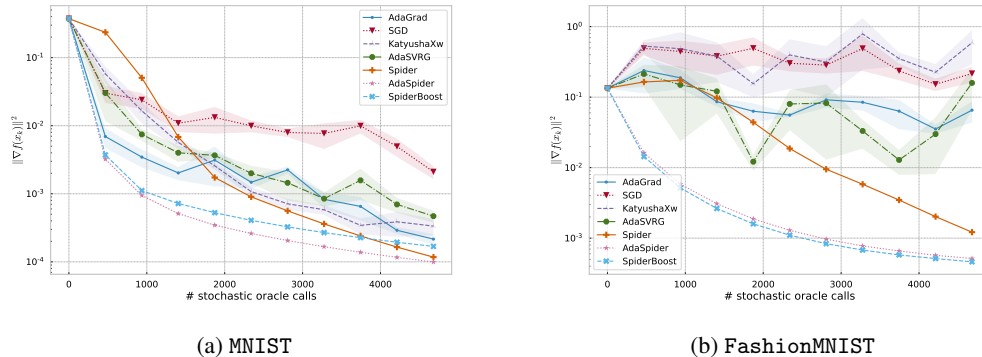

| (a) MNIST | (b) FashionMNIST |
|---|---|

Figure 2: Gradient norms throughout the epochs for image classification with neural networks (curves are averaged over 5 independent runs and the shaded region are the standard error).

Table 2: Algorithm parameters and test accuracies (average of 5 runs, in $\%$)

| | MNIST | | FashionMNIST | |
| | Batch Size = 32, $c_{\text{init}} = 0.03$ | | Batch Size = 128, $c_{\text{init}} = 0.01$ | |
| Algorithm | Parameters | Test Accuracy | Parameters | Test Accuracy |
|---|---|---|---|---|
| AdaGrad[16] | $\eta = 0.01$, $\epsilon = 10^{-4}$ | 97.86 | $\eta = 0.01$, $\epsilon = 10^{-4}$ | 86.19 |
| SGD[46] | $\eta = 0.01$ | 98.11 | $\eta = 0.01$ | 85.83 |
| KatyushaXw[3] | $\eta = 0.005$ | 97.93 | $\eta = 0.01$ | 86.27 |
| AdaSVRG[15] | $\eta = 0.1$ | 98.03 | $\eta = 0.1$ | 86.82 |
| Spider[19] | $\epsilon = 0.01$, $L = 100.0$, $n_0 = 1$ | 97.53 | $\epsilon = 0.01$, $L = 50.0$, $n_0 = 1$ | 82.22 |
| SpiderBoost[51, 42] | $L = 200$ | 97.01 | $L = 120$ | 84.42 |
| AdaSpider | $n = 60000$ | 97.49 | $n = 60000$ | 84.09 |

where $a$ is the input image. The training of the network consists of solving the following optimization problem: $\min_{x \in \mathbb{R}^d} \frac{1}{n} \sum_{i=1}^n \left( -b_i^\top \text{net}(x, a_i) + \text{logsumexp}(\text{net}(x, a_i)) \right)$. This is the default setup for doing image classification and we test our algorithm on two benchmark datasets : MNIST[30] and FashionMNIST[54]. We choose 3-layer fully connected network with dimensions $[28 * 28, 512, 512, 10]$. The activation function is the ELU [11].

**Initialization:** The initialization of the network is a crucial component to guarantee good performance. We find that a slight modification of the Kaiming Uniform initialization [22] improves the stability of the tested variance reduction schemes. For each layer in the network with $d_{in}$ inputs, the original method initializes the weights with independent uniform random variables with variance $\frac{1}{d_{in}}$. Our modification initializes with a smaller variance of $\frac{c_{\text{init}}}{d_{in}}$ with $c_{\text{init}}$ in the order of $0.01$. With this choice, we observed that fewer variance reductions schemes diverged, and standard algorithms like SGD and AdaGrad(for which the original method was tuned), were not penalized and performed well. This often overlooked initialization heuristic is the only "tuning" needed for AdaSpider.

**Observations:** We observe (Figure 2) that AdaSpider performs as well as other variance reduction methods in terms of minimizing the gradient norm. The key message here is that it does so without the need for extensive tuning. This diminished need for tuning is a welcome feature for deep learning optimization, but, often the true metric of interest is not the gradient norm, but the accuracy on unseen data, and on this metric variance reduction schemes are not yet competitive with simpler methods like SGD. With AdaSpider, the focus can go to finding the right initialization scheme and architecture to ensure good generalization without being distracted by other parameters like the step-size choice.

## Acknowledgments

This project has received funding from the European Research Council (ERC) under the European Union's Horizon 2020 research and innovation program (grant agreement n° 725594), the Swiss National Science Foundation (SNSF) under grant number 200021_205011 and Innosuisse.

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
