## A Omitted Proofs

We first state two technical lemmas that have been extensively used in the analysis of adaptive methods, the proofs of which can be found in [31].

**Lemma 6.** *Let a sequence of non-negative real numbers* $\alpha_1, \ldots, \alpha_T \geq 0$ *then*

$$\sqrt{\sum_{t=1}^{T} \alpha_t} \leq \sum_{t=1}^{T} \frac{\alpha_t}{\sqrt{\sum_{s=1}^{t} \alpha_s}}$$

**Lemma 7.** *Let a sequence of non-negative real numbers* $\alpha_1, \ldots, \alpha_T \geq 0$ *then*

$$\sum_{t=1}^{T} \frac{\alpha_t}{1 + \sum_{s=1}^{t} \alpha_s} \leq \log\left(1 + \sum_{t=1}^{T} \alpha_t\right)$$

**Lemma 1.** *Define the gradient estimator at point* $x$ *as* $\nabla_x := \nabla f_i(x) - \nabla f_i(y) + \nabla_y$ *where* $i$ *is sampled uniformly at random from* $\{1, \ldots, n\}$. *Then,*

$$\mathbb{E}\left[\|\nabla_x - \nabla f(x)\|^2\right] \leq L^2 \|x - y\|^2 + \mathbb{E}\left[\|\nabla_y - \nabla f(y)\|^2\right]$$

*Proof.*

$$
\begin{aligned}
\mathbb{E}\left[\|\nabla_x - \nabla f(x)\|^2\right] &= \mathbb{E}\left[\|\nabla f_i(x) - \nabla f_i(y) + \nabla_y - \nabla f(x)\|^2\right] \\
&= \mathbb{E}\left[\|\nabla f_i(x) - \nabla f_i(y) + \nabla_y - \nabla f(x) + \nabla f(y) - \nabla f(y)\|^2\right] \\
&= \mathbb{E}\left[\|\nabla f_i(x) - \nabla f_i(y) - \nabla f(x) + \nabla f(y)\|^2\right] \\
&\quad + 2 \cdot \mathbb{E}\left[\langle \nabla f_i(x) - \nabla f_i(y) - \nabla f(x) + \nabla f(y), \nabla_y - \nabla f(y)\rangle\right] \\
&\quad + \mathbb{E}\left[\|\nabla_y - \nabla f(y)\|^2\right]
\end{aligned}
$$

Notice that $\mathbb{E}\left[\nabla f_i(x) - \nabla f_i(y) - \nabla f(x) + \nabla f(y)\right] = 0$ due to the fact that $i$ is selected uniformly at random in $\{1, \ldots, n\}$ and thus $\mathbb{E}\left[\nabla f_i(x) - \nabla f_i(y)\right] = \nabla f(x) - \nabla f(y)$. The latter implies that,

$$
\begin{aligned}
\mathbb{E}\left[\|\nabla_x - \nabla f(x)\|^2\right] &= \mathbb{E}\left[\|\nabla f_i(x) - \nabla f_i(y) - \nabla f(x) + \nabla f(y)\|^2\right] + \mathbb{E}\left[\|\nabla_y - \nabla f(y)\|^2\right] \\
&\leq \mathbb{E}\left[\|\nabla f_i(x) - \nabla f_i(y)\|^2\right] + \mathbb{E}\left[\|\nabla_y - \nabla f(y)\|^2\right] \\
&\leq L^2 \cdot \mathbb{E}\left[\|x - y\|^2\right] + \mathbb{E}\left[\|\nabla_y - \nabla f(y)\|^2\right]
\end{aligned}
$$

where the first inequality follows by the identity $\mathbb{E}\left[\|X - \mathbb{E}\left[X\right]\|^2\right] = \mathbb{E}[\|X\|^2] - \|\mathbb{E}\left[X\right]\|^2$ and the second inequality by the smoothness of the function $f_i(x)$. ∎

**Lemma 2.** *Let* $x_0, x_1, \ldots, x_T$ *the points produced by Algorithm 1. Then,*

$$\sum_{t=0}^{T-1} \|\nabla_t\|^2 \leq \mathcal{O}\left(n^2 T^3 \cdot \left(\frac{L^2}{\beta_0^2} + \|\nabla f(x_0)\|^2\right)\right)$$

*Proof.* The selection of the step-size in Step 9 of Algorithm 1 implies that $\|x_{t+1} - x_t\|^2 = \|\gamma_t \nabla_t\|^2 \leq 1/\beta_0^2$. Due to the fact that every $n$ iterations a full-gradient computation is performed, the estimator $\nabla_t := \nabla f_{i_t}(x_t) - \nabla f_{i_t}(x_{t-1}) + \nabla_{t-1}$ can be equivalently written as

$$\nabla_t = \sum_{s=t-t \bmod n+1}^{t} (\nabla f_{i_s}(x_s) - \nabla f_{i_s}(x_{s-1})) + \nabla f(x_{t-t \bmod n})$$

As a result,

$$
\begin{aligned}
\|\nabla_t\|^2 &= \|\sum_{s=t-t \bmod n+1}^{t}(\nabla f_{i_s}(x_s) - \nabla f_{i_s}(x_{s-1})) + \nabla f(x_{t-t \bmod n})\|^2 \\
&\leq 2 \cdot \|\sum_{s=t-t \bmod n+1}^{t} \nabla f_{i_s}(x_s) - \nabla f_{i_s}(x_{s-1})\|^2 + 2 \cdot \|\nabla f(x_{t-t \bmod n})\|^2 \\
&\leq 2n \cdot \sum_{s=t-t \bmod n+1}^{t} \|\nabla f_{i_s}(x_s) - \nabla f_{i_s}(x_{s-1})\|^2 + 2 \cdot \|\nabla f(x_{t-t \bmod n})\|^2 \\
&\leq 2nL^2 \cdot \sum_{s=t-t \bmod n+1}^{t} \|x_s - x_{s-1}\|^2 + 2 \cdot \|\nabla f(x_{t-t \bmod n})\|^2 \\
&\leq \frac{2L^2 n^2}{\beta_0^2} + 2 \cdot \|\nabla f(x_{t-t \bmod n})\|^2
\end{aligned}
$$

Now, we want to upper bound $\|\nabla f(x_t)\|$ for any $t \leq T$ with respect to the initial gradient norm. Using again the step-size selection $\gamma_t$ we get,

$$
\begin{aligned}
\|\nabla f(x_t)\| &= \|\nabla f(x_t) - \nabla f(x_0) + \nabla f(x_0)\| \\
&\leq \|\nabla f(x_t) - \nabla f(x_0)\| + \|\nabla f(x_0)\| & \text{(Triangular inequality)} \\
&\leq L\|x_t - x_0\| + \|\nabla f(x_0)\| & \text{(Smoothness)} \\
&\leq L\|x_t - x_{t-1}\| + L\|x_{t-1} - x_0\| + \|\nabla f(x_0)\| & \text{(Triangular inequality)} \\
&\leq L\sum_{i=1}^{t} \|x_i - x_{i-1}\| + \|\nabla f(x_0)\| \\
&\leq \frac{Lt}{\beta_0} + \|\nabla f(x_0)\|
\end{aligned}
$$

As a result,

$$
\begin{aligned}
\sum_{t=0}^{T-1} \|\nabla_t\|^2 &\leq \sum_{t=0}^{T-1}\left(\frac{2L^2 n^2}{\beta_0^2} + 2 \cdot \|\nabla f(x_{t-t \bmod n})\|^2\right) \\
&\leq \frac{2L^2 n^2}{\beta_0^2} T + 2\sum_{t=0}^{T-1} \|\nabla f(x_t)\|^2 \\
&\leq \frac{2L^2 n^2}{\beta_0^2} T + 2\sum_{t=0}^{T-1}(\frac{Lt}{\beta_0} + \|\nabla f(x_0)\|)^2 \\
&= \frac{2L^2 n^2}{\beta_0^2} T + 2\sum_{t=0}^{T-1}\left(\frac{L^2 t^2}{\beta_0^2} + 2\frac{Lt}{\beta_0}\|\nabla f(x_0)\| + \|\nabla f(x_0)\|^2\right) \\
&\leq \frac{2L^2 n^2}{\beta_0^2} T + \frac{2L^2 T^3}{\beta_0^2} + \frac{4LT^2\|\nabla f(x_0)\|}{\beta_0} + 2T\|\nabla f(x_0)\|^2
\end{aligned}
$$

∎

**Lemma 3.** *Let $x_0, x_1 \ldots, x_T$ the sequence of points produced by Algorithm 1. Then,*

$$
\sum_{t=0}^{T-1} \mathbb{E}\left[\|\nabla_t - \nabla f(x_t)\|^2\right] \leq \mathcal{O}\left(\frac{Ln^{1/4}}{\beta_0}\sqrt{\log\left(1 + nT \cdot \left(\frac{L}{\beta_0 G_0} + \frac{\|\nabla f(x_0)\|}{G_0}\right)\right)}\right).
$$

*Proof.*

$$\mathbb{E}\left[\sum_{t=0}^{T-1}\|\nabla_t - \nabla f(x_t)\|\right] = \mathbb{E}\left[\sqrt{\left(\sum_{t=0}^{T-1}\|\nabla_t - \nabla f(x_t)\|\right)^2}\right]$$

$$\leq \sqrt{\mathbb{E}\left[\left(\sum_{t=0}^{T-1}\|\nabla_t - \nabla f(x_t)\|\right)^2\right]} \qquad \text{(Jensen's ineq.)}$$

$$\leq \sqrt{T}\cdot\sqrt{\mathbb{E}\left[\sum_{t=0}^{T-1}\|\nabla_t - \nabla f(x_t)\|^2\right]}$$

where the last inequality follows by the fact that $\|\sum_{t=0}^{T-1}y_t\|^2 \leq T\cdot\sum_{t=0}^{T-1}\|y_t\|^2$. By applying Lemma 1 to the estimator $\nabla_t := \nabla f_{i_t}(x_t) - \nabla f_{i_t}(x_{t-1}) + \nabla_{t-1}$ we get,

$$\begin{aligned}
\mathbb{E}\left[\|\nabla_t - \nabla f(x_t)\|^2\right] &\leq L^2\mathbb{E}\left[\|x_t - x_{t-1}\|^2\right] + \mathbb{E}\left[\|\nabla_{t-1} - \nabla f(x_{t-1})\|^2\right]\\
&\leq L^2\mathbb{E}\left[\gamma_{t-1}^2\|\nabla_{t-1}\|^2\right] + \mathbb{E}\left[\|\nabla_{t-1} - \nabla f(x_{t-1})\|^2\right]\\
&\leq L^2\mathbb{E}\left[\gamma_{t-1}^2\|\nabla_{t-1}\|^2\right] + \ldots + \mathbb{E}\left[\|\nabla_{t-(t \bmod n)} - \nabla f(x_{t-(t \bmod n)})\|^2\right]\\
&= \sum_{\tau = t-(t \bmod n)+1}^{t-1} L^2\mathbb{E}\left[\gamma_\tau^2\cdot\|\nabla_\tau\|^2\right]
\end{aligned}$$

where the last equality follows by the fact that $\mathbb{E}\left[\|\nabla_{t-(t \bmod n)} - \nabla f(x_{t-(t \bmod n)})\|^2\right] = 0$ (see Step 3 of Algorithm 1). As explained in Section 3, by a telescoping summation over $t$ we get that

$$\sum_{t=0}^{T-1}\mathbb{E}\left[\|\nabla_t - \nabla f(x_t)\|^2\right] \leq L^2 n\cdot\mathbb{E}\left[\sum_{t=0}^{T-1}\gamma_t^2\cdot\|\nabla_t\|^2\right].$$

Now as discussed in Section 3, using the step-size selection $\gamma_t$ of Algorithm 1 we can provide a bound on the total variance $\mathbb{E}\left[\|\nabla_t - \nabla f(x_t)\|^2\right]$

$$\begin{aligned}
\sum_{t=0}^{T-1}\mathbb{E}\left[\|\nabla_t - \nabla f(x_t)\|^2\right] &\leq L^2 n\,\mathbb{E}\left[\sum_{t=0}^{T-1}\gamma_t^2\cdot\|\nabla_t\|^2\right]\\
&= \frac{L^2\sqrt{n}}{\beta_0^2}\,\mathbb{E}\left[\sum_{t=0}^{T-1}\frac{\|\nabla_t\|^2}{\sqrt{n}G_0^2 + \sum_{s=0}^t\|\nabla_s\|^2}\right]\\
&\leq \frac{L^2\sqrt{n}}{\beta_0^2}\log\left(1 + \mathbb{E}\left[\sum_{t=0}^{T-1}\|\nabla_t\|^2/G_0^2\right]\right)\\
&\leq \frac{L^2\sqrt{n}}{\beta_0^2}\cdot\mathcal{O}\left(\log\left(1 + nT\cdot\left(\frac{L}{\beta_0 G_0} + \frac{\|\nabla f(x_0)\|}{G_0}\right)\right)\right)
\end{aligned}$$

where the second inequality follows by Lemma 7 and the third inequality by Lemma 2. Putting everything together we get

$$\frac{1}{T}\mathbb{E}\left[\sum_{t=0}^{T-1}\|\nabla_t - \nabla f(x_t)\|\right] \leq \frac{Ln^{1/4}}{\beta_0\sqrt{T}}\cdot\mathcal{O}\left(\sqrt{\log\left(1 + nT\cdot\left(\frac{L}{\beta_0 G_0} + \frac{\|\nabla f(x_0)\|}{G_0}\right)\right)}\right)$$

∎

**Lemma 4.** *Let $x_0, x_1, \ldots, x_{T-1}$ the sequence of points produced by Algorithm 1 and $\Delta_0 := f(x_0) - f^*$. Then,*

$$\mathbb{E}\left[\sum_{t=0}^{T-1}\|\nabla_t\|\right] \leq O\left(\Delta_0\cdot\beta_0 + G_0 + \frac{L}{\beta_0}\log\left(1 + nT\cdot\frac{L + \|\nabla f(x_0)\|}{G_0}\right) + \beta_0\cdot\mathbb{E}\left[\sum_{t=0}^{T-1}\gamma_t\|\nabla f(x_t) - \nabla_t\|^2\right]\right)\cdot n^{1/4}\sqrt{T}.$$

*Proof.* Let $\mathcal{F}_t$ denote the filtration at round $t$ i.e. all the random choices $\{i_0, \ldots, i_t\}$ and the initial point $x_0$. By the smoothness of $f$ we get that,

$$
\begin{aligned}
\mathbb{E}\left[f(x_{t+1}) \mid \mathcal{F}_t\right] &\leq \mathbb{E}\left[f(x_t) + \nabla f(x_t)^\top (x_{t+1} - x_t) + \frac{L}{2}\|x_t - x_{t+1}\|^2 \mid \mathcal{F}_t\right] \\
&= \mathbb{E}\left[f(x_t) - \gamma_t \nabla_t^\top \nabla f(x_t) + \frac{L}{2}\gamma_t^2 \|\nabla_t\|^2 \mid \mathcal{F}_t\right] \\
&\leq \mathbb{E}\left[f(x_t) + \frac{\gamma_t}{2}\|\nabla_t - \nabla f(x_t)\|^2 - \frac{\gamma_t}{2}(1 - L\gamma_t)\|\nabla_t\|^2 \mid \mathcal{F}_t\right]
\end{aligned}
$$

Thus,

$$
\mathbb{E}\left[\gamma_t \cdot \|\nabla_t\|^2\right] \leq 2\mathbb{E}\left[f(x_t) - f(x_{t+1})\right] + \mathbb{E}\left[L\gamma_t^2 \cdot \|\nabla_t\|^2\right] + \beta_0 \cdot \mathbb{E}\left[\gamma_t \cdot \|\nabla f(x_t) - \nabla_t\|^2\right].
$$

and by summing from $t = 0$ to $T - 1$ we get,

$$
\sum_{t=0}^{T-1} \mathbb{E}\left[\gamma_t \cdot \|\nabla_t\|^2\right] \leq 2\Delta_0 + \mathbb{E}\left[\sum_{t=0}^{T-1} L\gamma_t^2 \cdot \|\nabla_t\|^2\right] + \mathbb{E}\left[\sum_{t=0}^{T-1} \gamma_t \cdot \|\nabla f(x_t) - \nabla_t\|^2\right]
$$

Using the fact that $\gamma_t := n^{-1/4}\beta_0^{-1}\left(n^{1/2}G_0^2 + \sum_{s=0}^{t}\|\nabla_s\|^2\right)^{-1/2}$ on the second summation term,

$$
\begin{aligned}
\mathbb{E}\left[\sum_{t=0}^{T-1} \gamma_t \cdot \|\nabla_t\|^2\right] &\leq 2\Delta_0 + \mathbb{E}\left[\sum_{t=0}^{T-1} L\gamma_t^2 \cdot \|\nabla_t\|^2\right] + \mathbb{E}\left[\sum_{t=0}^{T-1} \gamma_t \cdot \|\nabla f(x_t) - \nabla_t\|^2\right] \\
&\leq 2\Delta_0 + \frac{L}{\beta_0^2} \cdot \mathbb{E}\left[\sum_{t=0}^{T-1} \frac{\|\nabla_t\|^2}{\sqrt{n}G_0^2 + \sum_{s=0}^{t}\|\nabla_t\|^2}\right] \\
&\quad + \mathbb{E}\left[\sum_{t=0}^{T-1} \gamma_t \cdot \|\nabla f(x_t) - \nabla_t\|^2\right] \\
&\leq 2\Delta_0 + \frac{L}{\beta_0^2} \cdot \mathcal{O}\left(\log\left(1 + nT \cdot \left(\frac{L}{\beta_0 G_0} + \frac{\|\nabla f(x_0)\|}{G_0}\right)\right)\right) + \mathbb{E}\left[\sum_{t=0}^{T-1} \gamma_t \cdot \|\nabla f(x_t) - \nabla_t\|^2\right]
\end{aligned}
$$

Using again the definition of the step-size $\gamma_t := n^{-1/4}\beta_0^{-1}\left(n^{1/2}G_0^2 + \sum_{s=0}^{t}\|\nabla_s\|^2\right)^{-1/2}$ we *lower bound the right-hand side* as follows,

$$
\begin{aligned}
\mathbb{E}\left[\sum_{t=0}^{T-1} \gamma_t \cdot \|\nabla_t\|^2\right] &\geq \mathbb{E}\left[\frac{\sum_{t=0}^{T-1}\|\nabla_t\|^2}{n^{1/4}\beta_0\sqrt{n^{1/2}G_0^2 + \sum_{t=0}^{T-1}\|\nabla_t\|^2}}\right] \\
&\geq \frac{G_0}{\beta_0} \cdot \mathbb{E}\left[\frac{\sum_{t=0}^{T-1}\|\nabla_t\|^2/\sqrt{n}G_0^2}{\sqrt{1 + \sum_{t=0}^{T-1}\|\nabla_t\|^2/\sqrt{n}G_0^2}}\right] \\
&\geq \frac{G_0}{\beta_0} \cdot \left(\mathbb{E}\left[\sqrt{\sum_{t=0}^{T-1}\|\nabla_t\|^2/\sqrt{n}G_0^2}\right] - 1\right) \\
&\geq \frac{1}{\beta_0 n^{1/4}\sqrt{T}}\mathbb{E}\left[\sum_{t=0}^{T-1}\|\nabla_t\|\right] - \frac{G_0}{\beta_0}
\end{aligned}
$$

By putting everything together we get,

$$
\begin{aligned}
\mathbb{E}\left[\sum_{t=0}^{T-1}\|\nabla_t\|\right] &\leq O\left(\Delta_0 \cdot \beta_0 + G_0 + \frac{L}{\beta_0}\log\left(1 + nT \cdot \left(\frac{L}{\beta_0 G_0} + \frac{\|\nabla f(x_0)\|}{G_0}\right)\right)\right) \\
&\quad + \mathcal{O}\left(\beta_0 \cdot \mathbb{E}\left[\sum_{t=0}^{T-1}\gamma_t\|\nabla f(x_t) - \nabla_t\|^2\right]\right) \cdot n^{1/4}\sqrt{T}.
\end{aligned}
$$

∎

**Lemma 5.** *Let $x_0, x_1, \ldots, x_{T-1}$ the sequence of points produced by Algorithm 1. Then,*

$$\mathbb{E}\left[\sum_{t=0}^{T-1} \gamma_t \cdot \|\nabla_t - \nabla f(x_t)\|^2\right] \le L^2 n \cdot \mathbb{E}\left[\sum_{t=0}^{T-1} \gamma_t^3 \cdot \|\nabla_t\|^2\right]$$

*Proof.* Let $\mathcal{F}_t$ denotes the filtration at round $t$ i.e. all the random choices $\{i_0, \ldots, i_t\}$ and the initial point $x_0$. At first notice that by the definition of $\gamma_t$ in Step 9 of Algorithm 1, $\gamma_t \le \gamma_{t-1}$, which we have to do to circumvent non-measurability issues, and thus

$$\mathbb{E}\left[\gamma_t \|\nabla_t - \nabla f(x_t)\|^2 \mid \mathcal{F}_{t-1}\right] \le \mathbb{E}\left[\gamma_{t-1} \cdot \|\nabla_t - \nabla f(x_t)\|^2 \mid \mathcal{F}_{t-1}\right]$$

Up next we derive a bound on $\mathbb{E}\left[\gamma_{t-1} \cdot \|\nabla_t - \nabla f(x_t)\|^2 \mid \mathcal{F}_{t-1}\right]$ using similar arguments with the ones used in Lemma 3. Notice that $\gamma_{t-1}$ is $\mathcal{F}_{t-1}$-measurable, hence we can treat in independent of the conditional expectation.

$$
\begin{aligned}
&\mathbb{E}\left[\gamma_{t-1}\|\nabla_t - \nabla f(x_t)\|^2 \mid \mathcal{F}_{t-1}\right] \\
=\ & \gamma_{t-1}\mathbb{E}\left[\|\nabla f_{i_t}(x_t) - \nabla f_{i_t}(x_{t-1}) - \nabla f(x_t) + \nabla f(x_{t-1}) + (\nabla_{t-1} - \nabla f(x_{t-1}))\|^2 \mid \mathcal{F}_{t-1}\right] \\
=\ & \gamma_{t-1}\mathbb{E}\left[\|\nabla f_{i_t}(x_t) - \nabla f_{i_t}(x_{t-1}) - \nabla f(x_t) + \nabla f(x_{t-1})\|^2 \mid \mathcal{F}_{t-1}\right] \\
+\ & \gamma_{t-1}\underbrace{\mathbb{E}\left[(\nabla f_{i_t}(x_t) - \nabla f_{i_t}(x_{t-1}) - \nabla f(x_t) + \nabla f(x_{t-1}))^\top(\nabla_{t-1} - \nabla f(x_{t-1})) \mid \mathcal{F}_{t-1}\right]}_{0} \\
+\ & \gamma_{t-1}\mathbb{E}\left[\|\nabla_{t-1} - \nabla f(x_{t-1})\|^2 \mid \mathcal{F}_{t-1}\right] \\
=\ & \gamma_{t-1}\mathbb{E}\left[\|\nabla f_{i_t}(x_t) - \nabla f_{i_t}(x_{t-1})\|^2 \mid \mathcal{F}_{t-1}\right] + \gamma_{t-1}\mathbb{E}\left[\|\nabla_{t-1} - \nabla f(x_{t-1})\|^2 \mid \mathcal{F}_{t-1}\right] \\
\le\ & L^2\gamma_{t-1}\mathbb{E}\left[\|x_t - x_{t-1}\|^2 \mid \mathcal{F}_{t-1}\right] + \gamma_{t-1}\mathbb{E}\left[\|\nabla_{t-1} - \nabla f(x_{t-1})\|^2 \mid \mathcal{F}_{t-1}\right] \\
=\ & L^2\gamma_{t-1}^3\mathbb{E}\left[\|\nabla_{t-1}\|^2 \mid \mathcal{F}_{t-1}\right] + \gamma_{t-1}\mathbb{E}\left[\|\nabla_{t-1} - \nabla f(x_{t-1})\|^2 \mid \mathcal{F}_{t-1}\right]
\end{aligned}
$$

Taking full expectation over all randomness and by the law of total expctation, we get that,

$$\mathbb{E}\left[\gamma_t\|\nabla_t - \nabla f(x_t)\|^2\right] \le L^2\mathbb{E}\left[\gamma_{t-1}^3\|\nabla_{t-1}\|^2\right] + \mathbb{E}\left[\gamma_{t-1}\|\nabla_{t-1} - \nabla f(x_{t-1})\|^2\right]$$

Due to the fact that $\mathbb{E}\left[\|\nabla_t - \nabla f(x_t)\|\right] = 0$ for $t \bmod n == 0$ we get that

$$\mathbb{E}\left[\gamma_t \cdot \|\nabla_t - \nabla f(x_t)\|^2\right] \le L^2\mathbb{E}\left[\sum_{s=t-t \bmod n}^{t-1} \gamma_s^3\|\nabla_s\|^2\right]$$

and thus

$$\mathbb{E}\left[\sum_{t=0}^{T-1} \gamma_t \cdot \|\nabla_t - \nabla f(x_t)\|^2\right] \le L^2 n \cdot \mathbb{E}\left[\sum_{t=0}^{T-1} \gamma_t^3\|\nabla_t\|^2\right]$$

∎

**Theorem 1.** *Let $x_0, x_1, \ldots, x_{T-1}$ be the sequence of points produced by Algorithm 1 in case $f(\cdot)$ is L-smooth. Let us also define $\Delta_0 := f(x_0) - f^*$. Then,*

$$\frac{1}{T}\sum_{t=0}^{T-1} \mathbb{E}\left[\|\nabla f(x_t)\|\right] \le O\left(n^{1/4} \cdot \frac{\Delta_0 \cdot \beta_0 + G_0 + L/\beta_0 + L^2/\beta_0^2 G_0}{\sqrt{T}} \cdot \log\left(1 + nT \cdot \left(\frac{L}{\beta_0 G_0} + \frac{\|\nabla f(x_0)\|}{G_0}\right)\right)\right)$$

*Overall, Algorithm 1 with $\beta_0 := 1$ and $G_0 := 1$ needs at most $\tilde{O}\left(n + \sqrt{n} \cdot \frac{\Delta_0^2 + L^4}{\epsilon^2}\right)$ oracle calls to reach an $\epsilon$-stationary point.*

*Proof of Theorem 1.* By the triangle inequality we get that

$$\mathbb{E}\left[\sum_{t=0}^{T-1} \|\nabla f(x_t)\|\right] \le \mathbb{E}\left[\sum_{t=0}^{T-1} \|\nabla_t\|\right] + \mathbb{E}\left[\sum_{t=0}^{T-1} \|\nabla f(x_t) - \nabla_t\|\right]$$

Using the bounds obtained in Lemma 3 and Lemma 4 we get that,

$$
\mathbb{E}\left[\sum_{t=0}^{T-1}\|\nabla f(x_t)\|\right] \leq \tilde{O}\left(\Delta_0 \cdot \beta_0 + G_0 + \frac{L}{\beta_0}\right) n^{1/4}\sqrt{T}
$$
$$
+ \quad \beta_0 \cdot \mathbb{E}\left[\sum_{t=0}^{T-1}\gamma_t\|\nabla f(x_t) - \nabla_t\|^2\right] n^{1/4}\sqrt{T}
$$

Then by Lemma 5 we get that,

$$
\mathbb{E}\left[\sum_{t=0}^{T-1}\|\nabla f(x_t)\|\right] \leq \tilde{O}\left(\Delta_0 \cdot \beta_0 + G_0 + \frac{L}{\beta_0}\right) n^{1/4}\sqrt{T}
$$
$$
+ \quad \beta_0 \cdot \underbrace{n^{5/4}\sqrt{T}L^2 \cdot \mathbb{E}\left[\sum_{t=0}^{T-1}\gamma_t^3\|\nabla_t\|^2\right]}_{(A)}
$$

Substituing the selection of $\gamma_t$ in term (A) we get,

$$
\beta_0\sqrt{T}L^2 \cdot \mathbb{E}\left[\sum_{t=0}^{T-1}n^{5/4}\gamma_t^3\|\nabla_t\|^2\right] = \frac{\sqrt{T}L^2}{\beta_0^2} \cdot \mathbb{E}\left[\sum_{t=0}^{T-1}\frac{n^{5/4}}{n^{3/4}\sqrt{n^{1/2}G_0^2 + \sum_{s=0}^{t}\|\nabla_t\|^2}} \cdot \frac{\|\nabla_t\|^2}{n^{1/2}G_0^2 + \sum_{s=0}^{t}\|\nabla_t\|^2}\right]
$$
$$
\leq \frac{\sqrt{T}L^2}{\beta_0^2 G_0} \cdot \mathbb{E}\left[\sum_{t=0}^{T-1}\frac{n^{5/4}}{n^{3/4}\sqrt{n^{1/2}}} \cdot \frac{\|\nabla_t\|^2/G_0^2}{n^{1/2} + \sum_{s=0}^{t}\|\nabla_t\|^2/G_0^2}\right]
$$
$$
\leq \frac{\sqrt{T}L^2}{\beta_0^2 G_0} \cdot n^{1/4} \cdot \mathbb{E}\left[\sum_{t=0}^{T-1}\frac{\|\nabla_t\|^2/G_0^2}{1 + \sum_{s=0}^{t}\|\nabla_t\|^2/G_0^2}\right]
$$
$$
\leq \frac{\sqrt{T}L^2}{\beta_0^2 G_0} \cdot n^{1/4} \cdot \mathcal{O}\left(\log\left(1 + nT \cdot \left(\frac{L}{\beta_0 G_0} + \frac{\|\nabla f(x_0)\|}{G_0}\right)\right)\right)
$$

where the forth inequality follows by Lemma 7 and the last by Lemma 2. Theorem 1 then follows by dividing both sides with $T$. ∎