# OpenReview forum: "Adaptive Stochastic Variance Reduction for Non-convex Finite-Sum Minimization"
_NeurIPS.cc/2022/Conference — NeurIPS 2022 Accept_

### Official Review · Reviewer_rV1y · 2022-07-12

**Rating:** 5
**Confidence:** 4
**Soundness:** 3 good
**Presentation:** 3 good
**Contribution:** 2 fair

**Summary:**

The paper proposed a adaptive variant of Spider, called AdaSpider, for solving non-convex and smooth finite-sum optimization problem with the complexity of $\tilde O(n + \sqrt{n}/\epsilon^2)$ of finding a first-order stationary point. In addition, the proposed AdaSpider is a parameter-free algorithm, which is pratical in real applications. Finally, the authors provided some experimental results on non-convex optimization and neural networks.

**Questions:**

It seems the analysis heavily follows the Spider paper, which is very straightforward. I can not see the advantage of convergence analysis in this paper. Besider, the experimental results reveal that Spiderboost is better than AdaSpider. Can you explain?

**Limitations:**

No.

**Strengths And Weaknesses:**

Pros: The orginization of this paper is good and its presentation is clear to me. The proposed AdaSpide is parameter-free, which is a significant contribution to the non-convex optimization community. The experiments with neural networks are studied.

Cons: The advantage of its convergence analysis over the standard one, i.e., Spider (Fang et. al), is not clear. Due to the additional $\log$ term, the convergence result of the proposed AdaSpider is even worse than Spider and Spiderboost (Wang et. al). The experimental results show that the proposed AdaSpider is worse than Spiderboost.

---

> ### Author Response · Authors · 2022-08-01
> **Initial Author Response**
>
> We thank the reviewer for their time and valuable comments.
>
> *Concerning the reviewer's comment on the convergence rate: (Copied from Reviewer 5sbc)* We note that adaptive methods typically admit sightly worse rates than their non-adaptive counterparts due to being oblivious to the parameter $L$ (e.g. Theorem $2$ in [30], Theorem 2.2 in [35], Theorem 2.1 in [50], Theorem 2 in [32] etc.).
>
> *Concerning the reviewer's comment on the convergence analysis of Adapsider:*
>
> We respectfully disagree that our analysis heavily follows by the analysis of Spider.
> - Our convergence analysis is based on coupling ideas of Spider with techniques developed in the context of adaptive methods.
> - The analysis of Spider heavily needs both the smoothness constant $L$ and the accuracy $\epsilon$ (on its step-size selection) to achieve the optimal $O(n + \sqrt{n}/\epsilon^2)$ gradient-complexity.
> - Overcoming the accuracy-dependent step-size of Spider is already a significant challenge. This is clearly indicated by the fact that SpiderBoost, which was latter proposed as an accuracy-independent counter-part of Spider, requires a way more involved analysis than Spider and Adaspider.
>
>  We believe that the use of adaptivity techniques to design both an accuracy and smoothness independent version of Spider with near-optimal rates and a neat proof should be considered as a merit of rather than a weakness.
>
> *(Copied from Reviwer 5sbc) Concerning the experimental performance of Adaspider:* We note the following points,
> - Our work is primarly theoretical and answers the question of existence of an adaptive VR method with optimal gradient complexity in the context of non-convex finite-sum optimization.
> - A recent paper answering the exact same theoretical question for the convex case was published in ICML 2022 (Adaptive Accelerated (Extra-)Gradient Methods with Variance Reduction, [Liu et al. 2022]).
> - It is true that Adaspider does not outperform a well-tuned version of SpiderBoost on FashionMNIST but it does so on MNIST. However it achieves similar performance without the need for tuning that is an expensive time-consuming step.
> - Adaspider vastly outperforms Spider indicating that adaptivity helps. As a result, an interesting future direction is the extension of our techniques in the design of an adaptive version of SpiderBoost achieving nearly the same rates with its non-adaptive counterpart but may perform significantly better in practice.

---

### Official Review · Reviewer_5sbc · 2022-07-12

**Rating:** 6
**Confidence:** 3
**Soundness:** 3 good
**Presentation:** 4 excellent
**Contribution:** 3 good

**Summary:**

This paper presents a convergence analysis of an adaptive variance-reduction method (AdaSpider), which combines Spider method with the global(norm) version of AdaGrad stepsizes, for the minimization of smooth functions with a finite time structure. In particular, they show that AdaSpider can get an $\epsilon$ stationary point with $\tilde{O}\left(n + \sqrt{n}/\epsilon^2 \right)$ oracle complexity in expectation. Numerical results are also conducted.

**Questions:**

I'm wondering if it is possible to extend the analysis to the convex setting. Will AdaSpider still yield an (almost) optimal oracle complexity?

**Strengths And Weaknesses:**

This paper is well written and easy to follow. As far as I know, it is the first parameter-free nonconvex variance reduction method. In this way, I tend to support accepting this paper. Yet I have the following concerns.

AdaSpider vs. Spider/Spiderboost: The advantage of AdaSpider is parameter-free, i.e. without prior knowledge of problem dependent parameter smoothness L/less tuning. However, theoretically, it has a worse dependency on the smoothness L as well as pays a logarithmic term compared to Spider. Also, though the method requires less tuning, it seems no better (sometimes worse) than the other variance reduction methods as shown in the experimental part.

Minor:  Line 83, the dependency on the smoothness should be $L^4$?

---

> ### Author Response · Authors · 2022-08-01
> **Initial Author Response**
>
> We thank the reviewer for their time and valuable comments.
>
> *Concerning the reviewer's comment on the convergence rate:* We note that adaptive methods typically admit sightly worse rates than their non-adaptive counterparts due to being oblivious to the parameter $L$ (e.g. Theorem~$2$ in [30], Theorem 2.2 in [35], Theorem 2.1 in [50], Theorem 2 in [32] etc.).
>
> *Concerning the experimental performance of Adaspider:* We note the following points,
> - Our work is primarly theoretical and answers the question of existence of an adaptive VR method with optimal gradient complexity in the context of non-convex finite-sum optimization.
> - A recent paper answering the exact same theoretical question for the convex case was published in ICML 2022 (Adaptive Accelerated (Extra-)Gradient Methods with Variance Reduction, [Liu et al. 2022]).
> - It is true that Adaspider does not outperform a well-tuned version of SpiderBoost on FashionMNIST but it does so on MNIST. However it achieves similar performance without the need for tuning that is an expensive time-consuming step.
> - Adaspider vastly outperforms Spider indicating that adaptivity helps. As a result, an interesting future direction is the extension of our techniques in the design of an adaptive version of SpiderBoost achieving nearly the same rates with its non-adaptive counterpart but may perform significantly better in practice.
>
> *Concerning the Reviewer's question on the convex setting:* We believe that one could establish the following rates for Adaspider on the convex setting, $O(n + \sqrt{n}/\epsilon)$ to reach an $\epsilon$-optimal point. However we remark that [Liu et al., 2022] provide an adaptive VR method with $O(n + \sqrt{n}/\sqrt{\epsilon})$ gradient complexity for the convex case.

---

### Official Review · Reviewer_kKhQ · 2022-07-12

**Rating:** 6
**Confidence:** 4
**Soundness:** 4 excellent
**Presentation:** 2 fair
**Contribution:** 3 good

**Summary:**

This paper considers the problem of finding a first-order stationary point of a non-convex function which decomposes into a finite sum of $L$-smooth functions. The main motivating prior work is the SPIDER algorithm, which is an stochastic variance-reduction algorithm with non-adaptive step-sizes which achieves an order-optimal convergence rate for this setting. However, this algorithm is only guaranteed this convergence rate when the step-sizes are carefully tuned to problem parameters (e.g., the accuracy $\epsilon$ and smoothness constant $L$). Since these parameters may not be known a-priori, and the smoothness constant in particular may be difficult to compute, the authors aim to design an algorithm which does not need to be tuned w.r.t. these parameters. They propose a variant of the SPIDER algorithm which uses adaptively-chosen step sizes, and show that, up to logarithmic factors, their method obtains order-optimal convergence without tuning the step-size with $\epsilon$ or $L$.

**Questions:**

I have outlined my main comments and questions in the **Limitations** section above. The main points that I would like to see addressed in the rebuttal are the following:

- A more detailed discussion of prior work, and its relation to your work, and some techniques you use (e.g., to removing bounded gradient assumptions) should be added to the paper. I would like to see an outline of how you plan to incorporate this discussion in your paper.
- I am particularly interested in hearing your responses to the points raised in _Technical novelty and limitations of results_, and would be interested to see if the results of your paper could be improved as suggested there, or if there are significant challenges in doing so.
- I think it is important to fix the issues regarding _Dimensional analysis_ in your paper, as these discrepancies that are currently in your results make quickly sanity-checking the results very difficult for the quick/casual reader.
- Regarding the missing lemmas/inequality explanations, I would like to see a discussion about this. Do you plan to incorporate these changes? Or have I missed something simple here?

**Please note** that my current score for this paper reflects the fact that there are a number of changes that should be addressed by the authors before it is ready for publication. I will happily upgrade my score if the concerns I have raised are satisfactorily addressed.


**Strengths And Weaknesses:**

**Strengths**:
This paper addresses an important problem of designing optimization algorithms which are guaranteed to converge even when the parameters of the objective function (namely, the smoothness constant) are not known, or are underestimated. The exposition of their algorithm, ideas, and analysis are well-written, and pretty concise and clear. Overall, I found this paper very accessible and easy to follow.

Moreover, except for some minor typos, some (inconsequential) weirdness w.r.t. dimensional analysis, and some expressions which are not fully explained, the results of this paper appear correct to me, after a thorough reading of the proofs.

**Weaknesses**:
There are a number of issues I came across in the paper (both technical, and in regards to prior work) that should be addressed. I will outline them below:

> _Discussion of prior work_

Your paper has missed a substantial amount of prior work which is very closely related to your paper. Indeed, the step-sizes that you consider are precisely the AdaGrad-Norm step sizes, introduced in “Less regret via online conditioning” (arXiv:1002.4862, 2010) in the online convex opt context, and analyzed further by “Scale-Free Algorithms for Online Linear
Optimization” (ALT’15) (again in the context of online), and later by, e.g., “AdaGrad stepsizes: Sharp Convergence over Nonconvex Landscapes” (ICML’19), “The Power of Adaptivity in SGD: Self-Tuning Step Sizes with Unbounded Gradients and Affine Variance” (COLT’22), and minor variations of this step-size by “On the Convergence of Stochastic Gradient Descent with Adaptive Stepsizes” (AISTATS’19), “A High Probability Analysis of Adaptive SGD
with Momentum” (ICML’20 workshop). There are many other papers in this line of work, and this is only a small sample.

Particularly related to your work is “The Power of Adaptivity in SGD: Self-Tuning Step Sizes with Unbounded Gradients and Affine Variance”, which deals with unbounded gradients, and guarantees their convergence rate without tuning the step size w.r.t. the smoothness constant $L$. In fact, Lemma 4 in your paper is essentially the same as Lemma 2 in that paper.

I believe your paper should include a discussion on the line of work of AdaGrad-Norm, since it is the algorithm you consider in this paper. Further, it would be good to draw connections to the other recent works which avoid the need for assumptions of bounded gradients.

Also, when referencing the AdaGrad algorithm, I would suggest also citing “Adaptive Bound Optimization for Online Convex Optimization” (COLT’10), which proposed the AdaGrad algorithm concurrently (and in the same conference) to the Duchi, Hazan, and Singer paper.

> _Technical novelty and limitations of results_:

I have some observations about the results of your paper in relation to those of prior work.

_The step-size choice_: While it is nice that your step-size choice does not need to be tuned w.r.t. the smoothness parameter $L$, this is not surprising to me, since AdaGrad-Norm is well-known to have this nice property (the earliest reference observing this property which I am aware of is “AdaGrad stepsizes: Sharp Convergence over Nonconvex Landscapes” (ICML’19), but I would not be surprised if it was observed in earlier work also).

In that prior work (as well as follow-up work to it), AdaGrad-Norm has an order-optimal convergence rate for _any_ choice of parameters. This seems to stand in contrast to your algorithm, where the step-size is tuned w.r.t. $n$ (which is, in a sense, a parameter which controls the variance, according to Lemma 5). It is not very intuitive to me why the step size in your setting should be tuned with this parameter (and, in particular, why the multiplicative factor of $1/n^{1/4}$ should be there). If $n = \Theta(T)$ (which I would assume is typically the regime of interest?), then this step size is typically smaller than $1/\sqrt{T}$, which I find somewhat surprising.

It would be very interesting, and perhaps a stronger result, if you could guarantee your convergence rate without tuning these parameters. Do you think this is possible? I noticed there seem to be a couple of places where there is some (small amount of) slack in your analysis (see my second comment in _Minor typos in some lemmas_). Do you think that by using this extra slack, your analysis could be improved to avoid this inverse multiplicative dependence on $n$ in the step-size? One possible place that I see where this (maybe?) could be improved is by more carefully upper-bounding in the second step of line 522, using the fact that for any $\lambda > 0$,
$$
-\gamma_t \nabla_t^\top (\nabla f(x_t) - \nabla_t)
\leq \frac{\lambda}{2} \lVert \nabla_t \rVert^2 + \frac{1}{2\lambda} \lVert \nabla f(x_t) - \nabla_t \rVert^2,
$$
and optimizing the choice of $\lambda$, perhaps as a function of $n$. In your analysis, you simply use $\lambda = 1$, but perhaps there is a better choice here that would improve your result?

_Convergence in “small-noise” regime_: Most results in stochastic first-order optimization (including all of the analyses of AdaGrad-Norm that I am aware of) obtain a faster convergence rate in the setting where the variance of the stochastic gradients is vanishingly small in $T$. In your setting, this perhaps could be viewed as many (or perhaps all) of the functions $f_i$ being the same (in this setting, your algorithm would obtain essentially a full estimate of the gradient for most time steps). I believe this should manifest itself as having a better dependence on $n$ in your convergence rate. Is there any way to quickly see this from Theorem 1? Or would a new, more refined analysis be needed to observe this phenomenon?

_Technical novelty_: The results in this paper have a simple and concise proof, which is nice! However, it seems (at least to me – perhaps I am missing something!) that the results seem to follow from a straightforward combination of techniques from the SPIDER paper, together with prior techniques used to analyze AdaGrad-Norm in the stochastic setting. In this sense, the results of this paper are not too surprising to me – I would expect that the AdaGrad-Norm step size applied to SPIDER would have a similar convergence rate without having to tune w.r.t. $L$. Were there other (perhaps subtle?) challenges that you faced in obtaining these results? It would be nice to highlight these, if so.

> _Proof of Theorem 1_

I think that the first inequality in proof of Theorem 1 (line 545) needs a bit more explanation. First, there is a small typo, as there should be a sum over all times on the left-hand side. Second, you should explicitly say first that Lemmas 5-7 imply that
$$
\frac{1}{n^{1/4} \sqrt{T}} \mathbb{E}[\lVert \nabla_t \rVert] \leq 2\Delta_0 + 1 + \frac{L \log(...)}{\sqrt{n}} + L^2 n \mathbb{E}[\gamma_t^3 \lVert \nabla_t\rVert^2].
$$
That inequality is easy to see by just combining those Lemmas. Then, you should show how to relate the LHS to the quantity you are interested in, using the observation that:
$$
\frac{1}{T} \sum_t \mathbb{E}[\lVert \nabla f(x_t) \rVert]
=
\frac{1}{T} \sum_t \mathbb{E}[\lVert \nabla_t + \nabla f(x_t) - \nabla_t \rVert]
\leq
\frac{1}{T} \sum_t \mathbb{E}[\lVert \nabla_t \rVert + \lVert \nabla f(x_t) - \nabla_t \rVert],
$$
together with the fact that, by Lemma 5,
$$
\frac{1}{T} \sum_t \mathbb{E}[\lVert \nabla f(x_t) - \nabla_t \rVert]
\leq
\frac{L n^{1/4} log(...)}{\sqrt{T}}.
$$
I am not sure if the above is very standard in this line of work (I suspect that it probably is), but since it was not written anywhere in this paper, I originally thought there was a serious mistake in the proof of Theorem 1. I would strongly suggest that you elaborate on that inequality from line 545 so that others don’t have to think as much about this :)

> _Dimensional analysis_:

There are many places in the paper where the dimensional analysis of your results does not quite check out. For instance, the quantity inside of the $\log$ in Theorem 1 is not unitless, since $n L T \lVert \nabla f(x_0) \rVert$ has units $f^2 / x^3$. For a detailed discussion on the problems with taking a logarithm of a quantity with units, you can see, e.g., the discussion in Appendix A of arxiv:2002.11259, “Dimensional Analysis in Statistical Modeling”. There are a number of other places in your paper where this problem also appears, e.g., Lemma 4, 5, 6, among other places.

I would like to clarify, before continuing, that I don’t think this invalidates your results (though it should be either fixed or clarified in your paper). Indeed, it seems that the source of this “strangeness” comes from your choice of step-size. Indeed, in the denominator, you add a “unitless” quantity, $n$, with a quantity with units of ($f^2 / x^2$), $\lVert \nabla f(x_t) \rVert^2$. Additionally, to make the units of the update check out (i.e., to make $\gamma_t \nabla_t$ have units $x$), the constant scaling the step-size ($1/n^{1/4}$ in your case) should have units of $x$. Thus, if one were to change the units of the domain $x$ or of the function $f$, then these terms in the step size would need to be scaled accordingly.

In order to track these units through your analysis (which, in my opinion, is one of the easiest ways to quickly sanity-check an optimization result), it would make things much easier if you included constants in your step size, scaling the multiplicative $1/n^{1/4}$ and additive $n^{1/2}$ terms. These constants would implicitly have the correct units, and would make verifying your result much easier on a quick read.

To conclude – I don’t think this issue creates any contradictory results for your guarantee. However, it makes reading some of the results a bit confusing, and throws some red flags (especially seeing a quantity with units inside a log). So I would suggest this issue be fixed throughout the paper.

> _Minor typos in some lemmas_:

I came across a number of minor errors/omissions (all inconsequential) in your technical results.

_In the proof of Lemma 4_,  on line 507, it is not clear to me why $L$ and $\lVert \nabla f(x_0) \rVert$ should be larger than $1$ w.l.o.g. When this is not true, then the last inequality in the following line is not true. Why not just replace the multiplication with an addition here?

_Not an error, but_ it seems to me (unless I’m mistaken) that, on the second line of the inequality following line 525 (inside proof of Lemma 6), that $L$ can actually be $L/\sqrt{n}$. Perhaps this makes some constants in your convergence rate slightly smaller? Not sure…

_Also in the proof of Lemma 6_, in the third inequality on line 527, I believe you are using the fact that:
$$
\sum_t \frac{a_t}{\sqrt{\sum_{s\leq t} a_s}} \geq \sqrt{\sum_t a_t}.
$$
This, of course, is a standard result which can be proven easily by induction. However, it is not written anywhere in your paper. Perhaps it should be included in the statement of Lemma 1? In fact, I didn’t find anywhere in your paper where Lemma 1 was used, and it seemed to me you only needed the lower bound. Perhaps you can just replace Lemma 1 with this bound?

_One last thing in proof of Lemma 6_, it seems that the $L^2$ inside the $\tilde{O}$ on line 528 should actually be replaced by $L/\sqrt{n}$. This makes some constants in later results slightly better.

_Small comment in proof of Lemma 7_, on lines 532-533, you say that “...which we have to do to circumvent non-measurability issues”. I was confused by this statement, as I don’t see why having a decreasing step size has anything to do with measurability… Perhaps this comment should be changed?

_Another small typo in proof of Lemma 7_, the 3rd equality on line 536 should be an inequality I believe (it’s equality by adding $- \gamma_{t-1}\lVert \nabla f(x_t) - \nabla f(x_{t-1})\rVert$). Not a big deal, just wanted to point this out.

---

> ### Author Response · Authors · 2022-08-01
> **Initial Author Response**
>
> We thank the reviewer for taking such a detailed look at our paper and the proof-checks of our results. We have already incorporated many of the reviewers comments on the missing inequalities and on the presentation (blue in the new supplementary material).
>
> Since the reviewer has raised many points and seems willing to engage in a further discussion, we provide some first brief answers to the main points so as to keep the subsequent discussion digestable.
> - *Step-Size selection:* As the reviewer mentions a first approach to the problem would be the selection of AdaGrad-Norm type step-sizes in the Spider method i.e.
>     $$\gamma_t = \frac{1}{\sqrt{1 +\sum_{s=0}^{t-1}||\nabla_t||^2}}.$$
> It is true that such a step-size adapts to the smoothness constant $L$, however the reviewer misses that this would lead to an overall $O(n/\epsilon^2))$ gradient-complexity, which is optimal w.r.t. $\epsilon$ but highly sub-optimal w.r.t. $n$ since the optimal gradient complexity for finite-sum minimization is $O(n + \sqrt{n}/\epsilon^2)$. Therefore, to achieve optimal gradient-complexity we select the step-size,
>     $$\gamma_t = \frac{1}{n^{1/4} \sqrt{n^{1/2} +\sum_{s=0}^{t-1}||\nabla_t||^2}}.$$
> The latter is a novel step-size selection in the adaptivity/VR literature and as the reviewer mentions seems unintuitive in the first sight.
>
> We additionally note to the reviewer that
> - The major challenge that we faced was to bound the variance in an amortized way while achieving the optimal $\sqrt{n}$ dependence.  More precisely, the basic problem is to simultaneously bound the overall expected norm of the estimator and the overall variance of the estimator (that are somewhat orthogonal goals) with the right $\sqrt{n}$ dependence.
> - Since $n$ denotes the number of components of $f$ while $T$ is the number of iterations the interesting regime is $T \gg n$. For example, a common practice in training neural networks is to make multiple passes over the training data.
> - *Small-variance regime:* This is an interesting question. To the best of our knowledge there are no variance-reduction methods (even beyond adaptivity) achieving better rates in \textit{small-variance instances}. We believe that the reason for this is that in the finite-sum case it is not clear how one should formally define small-variance instances. Nevertheless we believe that this is an interesting direction at which adaptivity can provide interesting answers.
> - Our results cannot be improved (up to logarithmic factors) due to the $\Omega(n + \sqrt{n}/\epsilon^2)$ lower bound of [Fang el. 2018].
> - *Dimensional Analysis:* We thank the reviewer for pointing out dimensional analysis as a proof-checking strategy. If the reviewer can provide us with some further clarifications/suggestions we are more than willing to incorporate the changes so as to make our claims more easily verifiable.
> To this end we have tried to incorporate the reviewer's comments by updating the manuscript as follows:
> As the reviewer writes in the proof of Lemma~4 we assume that $||\nabla f(x_0)|| , L \geq 1$ which is done for the sake exposition so that we present a simpler multiplicative form. As a result, instead of $nLT||\nabla f(x_0)||$ the right bound is $nT\cdot \max(L,1) \cdot \max(
> ||\nabla f(x_0)||,1)$. Thus the log-factors appearing in the proof admit the form $
> O(\log\left( nT\cdot \max(L,1) \cdot \max(
> ||\nabla f(x_0)||,1)\right))$ and thus the expression inside the $\log $ cannot be done less than $1$ even after rescaling of the variables. We have updated (in blue) the respective bounds in all the technical claims of the paper so as to facilitate sanity-check.
> - *Missing Related Work*: We thank the reviewer for the suggestion on the references. We have added discussion (lines 62-79) for these works in the updated version.
> - Many of the suggested papers study convergence properties of adaptive methods in the **general noise model** that in the non-convex case requires at least $\Omega(1/\epsilon^4)$ (or according to additional assumptions $\Omega(1/\epsilon^3)$) gradient-complexity. Instead in this work we focus on the **finite-sum structure**, $f(x):= (1/n) \sum_{i=1}^n f_i(x)$, for which the optimal gradient-complexity is $O( n + \sqrt{n}/\epsilon^2)$. In the updated version we have included an additional discussion (lines 62-80) comparing the two settings. We believe that these additional references add value to our write-up since they signify the interest on designing adaptive methods in the context of stochastic optimization. We have also included the suggested references using similar AdaGrad-type of step-size (lines 116-117).
> - Both Lemma $4$ (Lemma $2$ in the update version) in our work and Lemma $2$ in [Faw et al. 2022] (first part) state an immediate consequence of the smoothness of the function. We guess similar lemmas exist in various forms in the optimization literature. In any case, we have added a remark pointing their similarity (lines 125-132).

---

> > ### Author Response · Authors · 2022-08-02
> > **Update on Dimensional Analysis**
> >
> > In the last uploaded supplementary material (that also includes the updated main part) we have also incorporated the reviewer's suggestion on presenting the bounds so as to be compatible with the dimensional analysis. We have followed exactly the reviewer suggestion i.e.
> >
> > $$\gamma_t = \frac{1}{n^{1/4}\beta_0\cdot \sqrt{G_0^2\cdot n^{1/2} + \sum_{s=0}^{t-1}||\nabla_s||^2}}$$
> >
> > where the only role of $\beta_0$ and $G_0$ is to implicit fix the units ($\beta_0$ admits $x^{-1}$ units and $G_0$ admits $f/x$ units). In the final theorem $G_0$ and $\beta_0$ are set equal to $1$ and thus we derive the exact same complexity bounds as before but in the updated version all the technical claims are compatible with dimensional analysis.

---

> > > ### Comment · Reviewer_kKhQ · 2022-08-08
> > > **Follow-up on response**
> > >
> > > Thank you for the updates to the paper. I appreciate all of the changes you made to the paper in blue, and I think the paper is in a better state now (e.g., related work is now more comprehensive, and the missing Lemmas/explanations have been added). Some follow-up questions/comments:
> > >
> > > - _Step-size selection_: AdaGrad-Norm step-sizes are of the form $$\eta_t = \frac{\eta}{\sqrt{b_0^2 + \sum_{s=1}^t \lVert g_t \rVert^2}}$$ for any $\eta, b_0 > 0$ (following the notation in Ward, Wu, and Bottou '19), so yours is a special case by taking $1/\eta = b_0 = n^{1/4}$. This is why I view your algorithm as a special case of AdaGrad-Norm. So I would say the novelty in your step size is mainly how exactly to incorporate the structure of the function (i.e., $n$) into $\eta$ and $b_0$ in order to get the right dependence on $n$ in your convergence rate.
> > >
> > > - Going back to my earlier comment on Lemma 6 in the prior version/Lemma 4 in the updated version, it still seems to me that you are dropping a factor of $1/\sqrt{n}$ in the second inequality of line 579. Thus, the third term inside of the $\mathcal{O}(...)$ in the statement of Lemma 4 could actually be $\frac{L}{\sqrt{n}\beta_0}$ if I am not mistaken. In particular, there is an extra inverse dependence on $n$ that you are dropping (at least in one place) in your analysis. I bring this up because I am curious if it is really necessary to scale your step size by knowing $n$, or if it is possible to prove your same convergence rate for _any_ choice of the AdaGrad-Norm parameters $\eta$ and $b_0$. In most other analyses of AdaGrad-Norm, the order-optimal rate can be achieved without any tuning of $\eta$ and $b_0$, and I think it would be cool if this were the case in your analysis as well. Maybe the tuning of your step size w.r.t. $n$ is just an artifact of your analysis? Or can you give any evidence that tuning w.r.t. $n$ is necessary to obtain the correct rate.
> > >
> > > - Regarding the updates to the related works section (line 69), it comes across as if Ward et al introduced the AdaGrad-Norm algorithm. Perhaps the wording here should clarify that this was not the first work to study this algorithm.
> > >
> > > - Regarding the **general noise model** point on the related work -- for sure this is the case, but since you analyze the same algorithm in a different setting, I think this additional discussion is important to put your paper in context of the adaptive optimization literature. Thanks for adding this!
> > >
> > > - Regarding the dimensional analysis point -- thanks for adding this! I think it makes the results much easier to sanity-check, and everywhere I checked, dimensions seemed to check out.
> > >
> > > - Regarding Lemma 6 -- thanks for updating this to add the lower bound. However, the lemma doesn't seem to be actually referenced anywhere still. Perhaps you can add references to where exactly you use it (e.g., in Lemma 4). Also, I still do not see any place where the upper bound in Lemma 6 is used in your paper. Perhaps you can remove this part, and include only the lower bound?
> > >
> > > In summary, I would say I am now pretty happy with the state of the paper. My main outstanding question is whether there is hope to obtain the optimal convergence rate without tuning the step size w.r.t. $n$. At this time, it is not clear to me that it is necessary, or just an artifact of some sloppiness at some points of the analysis. It would be very interesting to me if you could recover the optimal rate without this specific parameter setting of AdaGrad-Norm.

---

> > > > ### Author Response · Authors · 2022-08-09
> > > > **Further follow-up**
> > > >
> > > > We thank the reviewer for the interesting discussion that helped us explore additional aspects that ameliorated our paper.
> > > >
> > > >
> > > > Given the degrees-of-freedom in AdaGrad-norm, we can indeed write our step-size within the class of AdaGrad-norm-type step-sizes. Hence, we state in the paper (see the current revision, lines 116-118) that our algorithm is a marriage of AdaGrad-norm and Spider with appropriately chosen constants for the step-size to achieve optimal scaling with $n$.
> > > >
> > > >
> > > > In what follows, we hope to resolve your main concern and would appreciate your feedback on our score.
> > > >
> > > >
> > > > Q: _In summary, I would say I am now pretty happy with the state of the paper. My main outstanding question is whether there is hope to obtain the optimal convergence rate without tuning the step size w.r.t. n. At this time, it is not clear to me that it is necessary, or just an artifact of some sloppiness at some points of the analysis. It would be very interesting to me if you could recover the optimal rate without this specific parameter setting of AdaGrad-Norm._
> > > >
> > > >
> > > > We believe that there is an expectation mismatch here. We already agree that combining the AdaGrad-Norm step-size with Spider would result in an adaptive method with optimal rate. However the issue is the scaling with the data dimension $n$, i.e. the optimal gradient-complexity $O(n + \sqrt{n}/\epsilon^2)$ is not achieved once $\eta$ and $b_0$ are set to some constants. The latter is indicated by the experiments that we subsequently present. More precisely, we compare the performance of Adaspider with the performance of Spider with Adagrad-Norm step-size with $\gamma = 1,b_0 =1$. This choice of parameters leads to similar performance for $n=10$ but the performance gap increases as the number of data points grows. We conduct experiments on subsets of the full MNIST dataset which is of size is $n=60000$. Our observations indicate that parameters $\gamma,b_0$ should be scaled with $n$ that the $n$-independent approach does not work.
> > > >
> > > > [Datasize 10](https://i.imgur.com/U2i6RQY.png), [Datasize 100](https://i.imgur.com/wPiXpy3.png), [Datasize 500](https://i.imgur.com/64PyV9N.png), [Datasize 1000](https://i.imgur.com/Va9u5aC.png), [Datasize 60000](https://i.imgur.com/PceMKt9.png) _(dashed lines correspond to AdaSpider and solid lines to Spider with Adagrad-Norm)_
> > > >
> > > > We thank the reviewer for reading our work in such detail to spot the fact that Lemma $6$ (Lemma $4$ in the updated version) admits a slight tighter bound. We remark that we chose the slightly looser bound in Lemma $6$ for the sake of simplicity. The inverse $\sqrt{n}$-dependence appears to the non-prevailing term of the Lemma $6$ (the prevailing term is $\mathrm{E}\left[\sum_t \gamma_t \cdot
> > > > ||\nabla_t - \nabla f(x_t)|| \right]$). As a result, a tighter analysis exploiting the inverse $\sqrt{n}$-dependence would only improve some constants on the final bound. We do not see how one could exploit the inverse $\sqrt{n}$-dependence to provide an $n$-independent step-size.
> > > >
> > > > We conclude by exhibiting the key-points in our analysis at which setting $\eta = n^{-1/4}$ and $b_0^2 = n^{1/2}$ plays a crucial role. We note that we do not see any way to overcome these obstacles once $\eta,b_0$ are universal constants.
> > > > -  Our analysis separately treats the terms $$ \mathrm{E}\left[\sum_t ||\nabla_t||\right],~~~\mathrm{E}\left[\sum_t ||\nabla_t - \nabla f(x_t)||\right].$$
> > > > - It is important to set $\eta = n^{-1/4}$ and $\beta_0^2 = n^{1/2}$ in order to establish that
> > > >     $$ \mathrm{E}\left[\sum_t ||\nabla_t||\right] \leq n^{1/4}\sqrt{T} \cdot \mathrm{E}\left[\sum_t \gamma_t ||\nabla_t||^2 \right]$$
> > > > Then using the form of the estimator we can establish that
> > > > $$ \mathrm{E}\left[\sum_t ||\nabla_t||\right] \leq n^{5/4}\sqrt{T} L^2 \cdot \mathrm{E}\left[\sum_t \gamma_t^3 ||\nabla_t||^2 \right]$$
> > > > -  For the second term we show that
> > > > $$\mathrm{E}\left[\sum_t ||\nabla_t - \nabla f(x_t)||\right] \leq \sqrt{T} \cdot \sqrt{L^2 n \cdot \mathrm{E}\left[\sum_t \gamma_t^2 ||\nabla_t||^2\right]}.$$
> > > > We note that the latter bound holds no matter what the form of the step-size.
> > > > -  Finally setting $\eta = n^{-1/4}$ and $b_0^2 = n^{1/2}$ makes both terms,
> > > >     -  $n^{5/4}\mathrm{E}\left[\sum_t \gamma_t^3 ||\nabla_t||^2 \right] = \mathcal{O}\left(n^{1/4}\right)$
> > > >
> > > >     - $\sqrt{n \cdot \mathrm{E}\left[\sum_t \gamma_t^2
> > > >     ||\nabla_t||^2\right]}= \mathcal{O}\left(n^{1/4}\right)$
> > > >
> > > > and implies that $ \mathrm{E}\left[\sum_t||\nabla f(x_t) || \right]\leq n^{1/4}/\sqrt{T}$. Thus, at most $O(n + \sqrt{n}/{\epsilon^2})$ gradient-calls are needed.
> > > >
> > > >
> > > > *Minor Comments*
> > > > -  In the revisited version we modified the wording about AdaGrad-Norm so that we avoid any confusion.
> > > > - Thank you for pointing this. In fact we only use the lower bound so in the revisited version we removed the upper bound from the lemma.

---

> > > > > ### Comment · Reviewer_kKhQ · 2022-08-09
> > > > > **Follow-up on response**
> > > > >
> > > > > Hi, thanks for the additional details! And thanks for the experiments for increasing dataset size, this helps a bit with intuition. So I guess one reason (perhaps not the only one, as you mention) that tuning the step size is important is to make sure that the variance has the proper $n$ dependence. As you write on line 567, variance will scale with $n$ if you do not scale down the step size as you do in your paper. And clearly getting proper variance scaling is important for variance reduction methods, so I guess it somewhat makes sense that there is some tuning of AdaGrad-Norm required.
> > > > >
> > > > > For a future version of your paper, I think it might be useful to add some of this intuition on why tuning the step size is crucial here, as it is typically not the case for the AdaGrad-Norm algorithms.
> > > > >
> > > > > I upgraded my score earlier today to 6/Weak accept. I think the paper is technically solid and well-written, and is a good addition to the growing body of literature showcasing the useful properties of adaptive optimization algorithms. However, these are the results that I would expect one would obtain from combining AdaGrad-Norm with the Spider algorithm, given the known convergence rate of Spider, and the prior work on AdaGrad-Norm showing "self-tuning" properties of the step-size w.r.t. problem parameters. Thus, while the techniques are not too surprising (and if there is reason to be surprised, or major challenges compared to prior work, it would be good to highlight this in the next version of your paper!), I view the result as nice and interesting (particularly the fact that tuning w.r.t. $n$ seems crucial).

---

### Author Response · Authors · 2022-08-08
**Thank you for your work. Any more feedback for us?**

Dear Reviewers,

Thank you very much for taking the time to review our paper, for your comments and for your helpful suggestions. If possible, we would appreciate some feedback on whether our responses have sufficiently addressed your questions/concerns.

Thank you again.

---

### Meta-Review · Area_Chair_7LJx · 2022-08-26

**Recommendation:** Accept
**Confidence:** Less certain

**Metareview:**

After discussion with the authors, all reviewers are satisfied with the technical quality of the paper. Achieving smoothness adaptivity in variance reduction is a significant result.

**Award:**

No

---

### Decision · Program_Chairs · 2022-09-14

Accept